# A role and mechanism for redox sensing by SENP1 in β-cell responses to high fat feeding

Haopeng Lin[1,2,3,9], Kunimasa Suzuki[1,2,9], Nancy Smith[1,2], Xi Li[4], Lisa Nalbach[5,6], Sonia Fuentes[4], Aliya F. Spigelman [1,2], Xiao-Qing Dai[1,2], Austin Bautista[1,2], Mourad Ferdaoussi[7], Saloni Aggarwal[8], Andrew R. Pepper [8], Leticia P. Roma [6], Emmanuel Ampofo[5], Wen-hong Li [4] & Patrick E. MacDonald [1,2] ✉

Pancreatic β-cells respond to metabolic stress by upregulating insulin secretion, however the underlying mechanisms remain unclear. Here we show, in β-cells from overweight humans without diabetes and mice fed a high-fat diet for 2 days, insulin exocytosis and secretion are enhanced without increased $Ca^{2+}$ influx. RNA-seq of sorted β-cells suggests altered metabolic pathways early following high fat diet, where we find increased basal oxygen consumption and proton leak, but a more reduced cytosolic redox state. Increased β-cell exocytosis after 2-day high fat diet is dependent on this reduced intracellular redox state and requires the sentrin-specific SUMO-protease-1. Mice with either pancreas- or β-cell-specific deletion of this fail to up-regulate exocytosis and become rapidly glucose intolerant after 2-day high fat diet. Mechanistically, redox-sensing by the SUMO-protease requires a thiol group at C535 which together with $Zn^+$-binding suppresses basal protease activity and unrestrained β-cell exocytosis, and increases enzyme sensitivity to regulation by redox signals.

Type 2 diabetes (T2D) occurs when insulin secretion from pancreatic β-cells fails to meet peripheral demand, which is increased with insulin resistance often coincident with obesity[1]. While obesity and insulin resistance are major risk factors for T2D, most individuals with obesity maintain normoglycemia as insulin secretion is increased adequately through the up-regulation of β-cell function and mass[2–4], although the relative contribution of each remains unclear[1]. Up-regulation of β-cell insulin secretory capacity precedes increases in islet mass in mice on a high-fat diet (HFD), and at early time points β-cell functional changes likely outweigh structural changes in increased insulin responses[5,6]. Loss of β-cell functional up-regulation may contribute to early progression and deterioration to T2D even with intact β-cell mass[7].

Upon short-term starvation, insulin secretion is attenuated with a shift from glucose metabolism to fatty acid oxidation[8]. On the other hand, overnutrition leads to a compensatory increase of insulin secretion with adaptations in β-cell stimulus-secretion coupling. These include enhanced intracellular $[Ca^{2+}]_i$ responses and up-regulation of metabolic coupling factors that potentiate $Ca^{2+}$ efficacy and amplify insulin secretion[9–11]. In a longitudinal study measuring parallel β-cell insulin secretion and $[Ca^{2+}]_i$ responses, the efficacy of $Ca^{2+}$-induced insulin secretion was enhanced via Epac signaling in the early stages of prediabetes[5]. This may be accompanied by up-regulation of reducing equivalents[12–14] and the exocytotic machinery per se in prediabetes[15], subsequently followed by a reduced expression in T2D[16].

[1]Department of Pharmacology, University of Alberta, Edmonton, AB T6G 2E1, Canada. [2]Alberta Diabetes Institute, University of Alberta, Edmonton, AB T6G 2E1, Canada. [3]Guangzhou Laboratory, Guangzhou 510005 Guangdong, China. [4]Departments of Cell Biology and Biochemistry, University of Texas Southwestern Medical Center, 6000 Harry Hines Blvd., Dallas, TX 75390-9039, USA. [5]Institute for Clinical & Experimental Surgery, Saarland University, Homburg/Saar, Germany. [6]Biophysics Department, Center for Human and Molecular Biology, Saarland University, Homburg/Saar, Germany. [7]Faculty Saint-Jean, University of Alberta, Edmonton, AB T6G 2E1, Canada. [8]Department of Surgery, University of Alberta, Edmonton, AB T6G 2E1, Canada. [9]These authors contributed equally: Haopeng Lin, Kunimasa Suzuki. ✉e-mail: pmacdonald@ualberta.ca

SUMOylation is the posttranslational conjugation of small ubiquitin-like modifier (SUMO) peptides to target proteins, which can be reversed by the sentrin-specific proteases (SENPs) such as SENP1 which in β-cells localizes near sites of docked insulin granules[17]. The activity of SENP1 is subject to redox regulation that may involve direct oxidation-reduction of cysteine residues near the enzyme catalytic site[18], although the exact structure-function relationship of SENP1 redox-sensing remains unclear. Whether 'permissive' or 'regulatory' to insulin secretion, cytosolic reducing signals help to maintain a robust pool of releasable secretory granules[19,20]. These signals act in part via SENP1, which modifies several exocytotic and exocytosis-related proteins to facilitate insulin granule priming without significantly affecting $Ca^{2+}$ entry[17,21]. Indeed, glucose stimulates deSUMOylation of the exocytotic proteins synaptotagmin VII and tomosyn 1 as a mechanism to regulate exocytotitc protein complex formation[21,22]. Several additional exocytotic proteins also appear regulated by SUMOylation[23]. In a chronic overnutrition model, glucose-stimulated insulin secretion (GSIS) is impaired, in part due to partial inactivation of SENP1 by oxidative stress[24,25]. In contrast, it remains unknown whether, at an early compensatory or prediabetic stage, altered redox state and SENP1 activity affect β-cell functional compensation and glucose homeostasis.

We therefore assessed the role for redox signaling in the up-regulation of β-cell exocytosis that occurs early upon high fat feeding in mice, and the structure-function relationship of SENP1 redox sensing. A reduced β-cell redox, commensurate with metabolic rewiring and an increased basal oxygen consumption, enhances exocytotic capacity via a mechanism that requires SENP1. In pancreas- and β-cell–specific SENP1 knockout mice, we demonstrate the requirement for SENP1 in the maintenance of glucose homeostasis within 2 days of high fat feeding via upregulated insulin secretion. Finally, we show that C535 near the catalytic site, together with an interaction with $Zn^{2+}$, is required for redox-regulation of enzyme activity and control of β-cell exocytosis.

## Results

### Higher β-cell secretory capacity from overweight humans and short-term HFD-fed mice

Islets from non-diabetic human donors (Supplementary Data 1) of 21–45 years of age and BMI >25 exhibited significantly elevated glucose-stimulated insulin secretion (GSIS) compared to similarly aged donors with BMI <25. This greater functionality was absent in T2D (BMI >25) (Fig. 1A). Depolarization-induced β-cell exocytosis measured by patch-clamp electrophysiology is amplified by glucose[17]. Intriguingly, β-cells from the donors with BMI >25 did not show any difference in depolarization-induced exocytosis compared to BMI <25 at 5- or 10-mM glucose. Instead, the exocytotic response was elevated in the non-diabetic donors with BMI >25 at 1 mM glucose (Fig. 1B). This means that the β-cells of higher-BMI donors upregulate their capacity to mount an exocytotic response at low glucose, to the same upper limit observed when the cells are pre-cultured at elevated glucose. That capacity does not translate to more insulin secretion at low glucose, however, but instead to an increased secretory response upon subsequent stimulation with glucose at levels sufficient to elicit an electrical and $Ca^{2+}$ response.

As with insulin secretion, this greater exocytotic capacity at 1 mM glucose was missing in β-cells of donors with T2D and BMI >25 (Fig. 1B). These differences were not as clear in islets and β-cells from female donors (Supplementary Fig. S1) or older donors (age >45; Fig. 1C, D), the latter perhaps being confounded by increasing stress responses, β-cell immaturity phenotypes, and generally reduced insulin secretion seen with age[26]. Notably however, insulin secretion and β-cell exocytosis were still lower in T2D (Supplementary Fig. S1, Fig. 1C, D).

To investigate β-cell functional changes that occur early under metabolic stress, we fed male C57BL/6NCrl mice either a chow diet (CD) or HFD for 2 days. Mice fed a 2-day HFD maintained normal glucose tolerance in an intraperitoneal glucose tolerance test (IPGTT) (Fig. 1E), along with increased plasma insulin (Fig. 1F). GSIS was increased from islets of 2-day HFD mice (Fig. 1G), without any effect on insulin content (Fig. 1H). Comparatively, after a 4-week HFD, mice were glucose intolerant (Supplementary Fig. S2A) with increased plasma insulin (Supplementary Fig. S2B), increased islet insulin content, and enhanced in vitro insulin secretion (Supplementary Fig. S2C–E). In β-cells from 2-day (Fig. 1I) and 4-week (Supplementary Fig. S2F) HFD-fed mice, exocytotic responses are increased at low glucose, to the same level as at elevated glucose. At 10 mM glucose there were no differences in voltage-activated $Ca^{2+}$ entry (Fig. 1J) and intracellular $Ca^{2+}$ responses (Fig. 1K; Supplementary Fig. S2G), consistent with previous demonstrations of 'increased $Ca^{2+}$ efficacy' under short-term HFD[5]. Indeed, under conditions that 'clamp' intracellular $Ca^{2+}$ responses to assess glucose-dependent amplification of insulin secretion[27], islets from 4-week HFD fed mice exhibited enhanced KCl-stimulated insulin secretion at low glucose (Supplementary Fig. S2E).

### Shifts in metabolic pathway expression after short-term HFD

To gain insight into the mechanism contributing to increased insulin secretion following short-term HFD[28], we performed RNA sequencing on FACS-sorted β-cells from 2-day CD and HFD mice (Supplementary Data 2). We found 213 differentially expressed (DE) genes (Fig. 2A). Enrichment analysis identified pathways (PW) such as histone methylation[29], MAPK signaling, cholesterol biosynthesis, insulin receptor signaling, and ER-associated misfolded protein response (Fig. 2B; Supplementary Data 3)[29,30]. Histone methylation, the most significantly enriched pathway, can interact or overlap with genes in other enriched pathways, such as MAPK signaling, consistent with the ability of methylation to adapt insulin secretion through MAPK and regulation of glucose metabolism[29]. All DE genes were submitted to the STRING database[31] for protein-protein interaction. Gene set enrichment analysis (GSEA) showed that up-regulated genes were enriched for glycolysis (*Tpi1*, *Pgk1*, *Pkm*…) and oxidative phosphorylation (Uqcrq, Vcp, Atp5g3…) pathways (Fig. 2C; Supplementary Data 4). Interestingly, the cholesterol biosynthesis pathway (*Fdft1*, *Hsd17b7*, *Msmo*…) was significantly down-regulated (Fig. 2C, D). Cholesterol biosynthesis relies heavily on NADPH consumption[32], and its inhibition may enhance insulin secretion by increasing NADPH[33]. Therefore, the downregulation of cholesterol biosynthesis genes could spare cytosolic NADPH, which might be built up by increased glucose metabolism, for the amplification of insulin release (Fig. 2D).

### Increased β-cell exocytosis after 2-day HFD requires a reducing signal

Reducing signals ensure a robust pool of releasable insulin granules[17]. Islets from 2-day HFD-fed mice appeared to have higher basal $O_2$ consumption and proton-leak measured by Seahorse assay (Fig. 3A). We confirmed an increased basal $O_2$ consumption rate in 2-day HFD islets by fluorescence lifetime measurement (Fig. 3B). In islets of mice expressing the cytosolic redox sensor Cyto-roGFP2-Orp1[34], we found that the cytosol in β-cells from 2-day HFD-fed mice was more reduced compared to control mice fed CD (Fig. 3C). A reduced redox state appears required for enhanced β-cell exocytosis since this was recapitulated in β-cells of CD-fed mice upon direct intracellular dialysis of reduced glutathione (GSH), which could not increase exocytosis further in β-cells of HFD-fed mice (Fig. 3D) and was reversed by direct intracellular application of $H_2O_2$ (Fig. 3E). The redox enzyme glutaredoxin 1 (GRX1) is required for GSH- and NADPH-dependent facilitation of insulin exocytosis[35,36], and we showed this likely acts via control of SENP1 activity (Fig. 3F)[17]. We observed an upregulation of *Senp1* expression by qPCR in islets after 2-day HFD, which was lost by 8-weeks

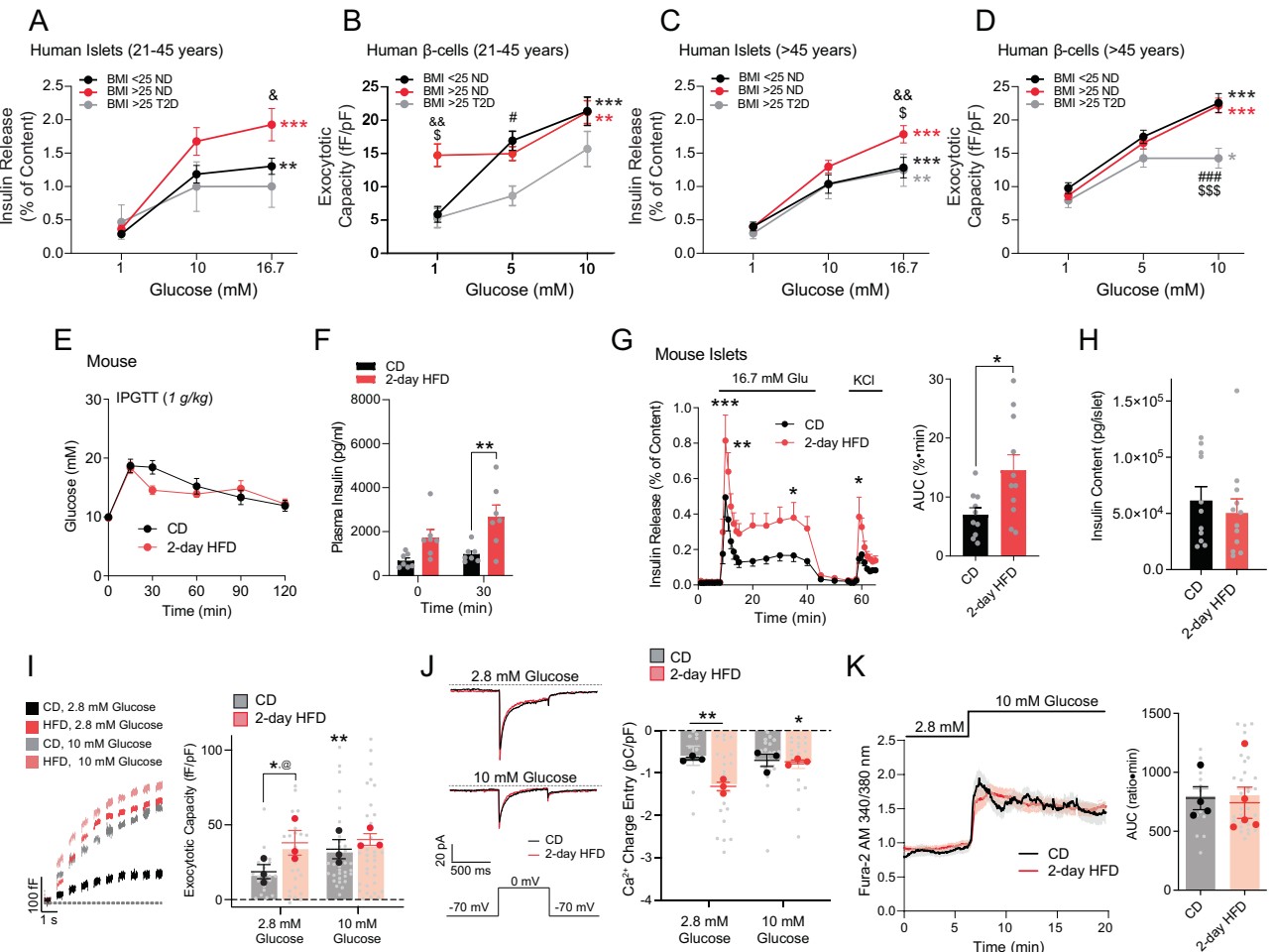

**Fig. 1 | Higher insulin secretion and β-cell exocytosis from islets of overweight donors and 2-day HFD mice. A** Insulin secretion from islets of young (21–45 years) donors at 1 ($n$ = 27, 47, 5), 10 ($n$ = 27, 41, 4) and 16.7 mM ($n$ = 27, 47, 5) glucose. Effect of glucose (**$p$ = 0.0010; ***$p$ = 1 × 10^{−11}) and vs BMI < 25 (&$p$ = 0.034). **B** Exocytosis from β-cells of young (21–45) donors at 1 ($n$ = 55, 120, 22), 5 ($n$ = 130, 236, 36) and 10 mM ($n$ = 88, 151, 21) glucose. Effect of glucose (**$p$ = 0.0056; ***$p$ = 4.6 × 10^{−7}), vs BMI < 25 (&&$p$ = 0.0039), vs T2D BMI > 25 ($$p$ = 0.046; #$p$ = 0.028). **C** Insulin secretion from islets of older (>45 years) donors 1 ($n$ = 61, 111, 21), 10 ($n$ = 40, 88, 18) and 16.7 mM ($n$ = 55, 110, 21) glucose. Effect of glucose (**$p$ = 0.0029; ***$p$ = 2.3 × 10^{−10}, 2.8 × 10^{−7}), vs BMI < 25 (&&$p$ = 0.0047), vs T2D BMI > 25 ($$p$ = 0.040). **D** Exocytosis from β-cells of older (>45 years) donors at 1 ($n$ = 164, 285, 78), 5 ($n$ = 310, 474, 112) and 10 mM ($n$ = 173, 364, 112) glucose. Effect of glucose (*$p$ = 0.031; ***$p$ = 7.9 × 10^{−11}; 5.8 × 10^{−11}), vs BMI > 25 (###$p$ = 1.8 × 10^{−4}), vs BMI > 25 ($$$p$ = 5.3 × 10^{−5}). Data in **A**–**E** compared by two-way ANOVA followed by Tukey post-test. See Supplementary

Fig. 1 for breakdown by sex and donor. **E** IPGTT of mice after CD and 2-day HFD ($n$ = 14, 14). **F** Plasma insulin during IPGTT ($n$ = 7, 7; **$p$ = 0.0077). **G, H** Insulin secretion (panel **G**; $n$ = 10, 11; ***$p$ = 5.1 × 10^{−5}; **$p$ = 0.0015, *$p$ = 0.041, 0.011) and content (panel **H**; $n$ = 11, 12). AUC – area under the curve (*$p$ = 0.019). **I** Representative traces (left), and average total responses, of β-cell exocytosis at 2.8 and 10 mM glucose ($n$ = 21, 26, 32, 33 cells, @$p$ = 0.017, from 3 pairs of mice, *$p$ = 0.015, 0.040). **J** Representative traces, and average Ca²⁺ currents, of β-cells at 2.8 and 10 mM glucose ($n$ = 9, 22, 19, 19 cells, *$p$ = 0.023, 0.026, from 3 pairs of mice). **K** Single cell [Ca²⁺]ᵢ response ($n$ = 16, 34 cells from 4, 5 mice). Data are mean ± SEM, compared with student unpaired Student's $t$ test (two-sided; panels **G**, **H**, **K**) or two-way ANOVA followed by Tukey post-test (panels **F**, **G**, **I**, **J**), uncorrected Fisher's LSD (panel **J**) or RM two-way ANOVA with matching and Bonferroni post-test (panel **I**). Source data are provided as a Source Data file.

(Fig. 3G). Direct intracellular dialysis of active SENP1 catalytic domain increases exocytosis in β-cells from CD-fed mice but, like GSH, cannot increase exocytosis further in the β-cells of HFD-fed mice (Fig. 3H).

In pSENP1-KO mice[24], males become rapidly glucose intolerant compared with control littermates, with significant IP and oral glucose intolerance after 2-day HFD (Fig. 4A–D). Loss of β-cell SENP1 prevents the up-regulation of exocytosis after either a 2-day (Fig. 4E) or 4-week (Fig. 4F) HFD. Similar to the pSENP1-KO line, male βSENP1-KO mice[24] become rapidly intolerant of IP glucose compared with control littermates upon 2-day HFD (Fig. 4G, H), with an impaired plasma insulin response (Fig. 4I). Consistent with our observation in human islets (Supplementary Fig. S1), these responses were less obvious in female mice from both the pSENP1-KO and βSENP1-KO lines (Supplementary Figs. S3 and S4), which are more resistant to the development of insulin resistance upon HFD[37]. Also, after 4-week HFD, oral glucose intolerance became more prominent in the βSENP1-KO females

(Supplementary Fig. S4I) and males (Fig. 4J–L, Supplementary Fig. S5), although again the females were generally more insensitive, consistent with our recent report at 8 weeks of HFD in both the pSENP1- and βSENP1-KO models[24].

## Redox regulation of exocytosis requires SENP1 C535

SENP1, a cysteine protease, is subject to redox regulation via a thiol side chain of the catalytic cysteine C603 or possibly other cysteines[18]. C603 is the most conserved cysteine across SENP isoforms and species, while C535 is the least conserved near the active site (Fig. 5A). Although redox regulation of intra-molecular disulfide bonds may control cysteine protease activity[38], the key cysteines of the SENP1 catalytic domain appear too distant from the catalytic C603 to facilitate such regulation[39] (Supplementary Fig. S6A). We therefore generated a series of recombinant SENP1 catalytic domain proteins with cysteine-to-serine substitutions and measured SUMO-protease activity

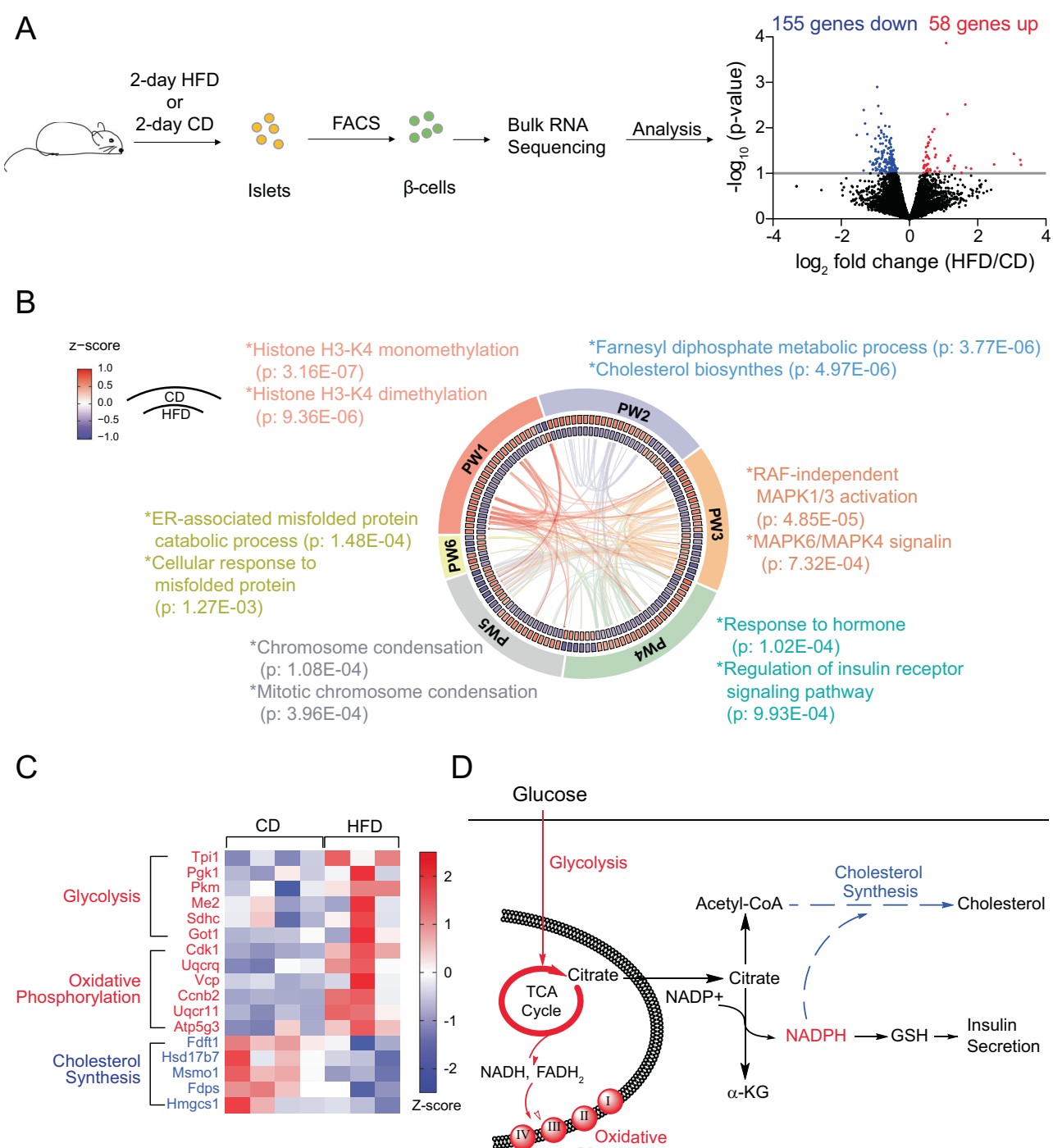

**Fig. 2 | RNA sequencing of purified β-cells following 2-day HFD. A** β-cells from CD and 2-day HFD were isolated through fluorescence activated cell sorting (FACS) for RNA sequencing. 213 genes were identified differentially expressed (DE) genes after 2-day HFD (213 genes, $n = 4$ and 3 mice). No adjustments were made for multiple comparisons. **B** All DE genes were submitted to Metascape for functional enrichment analysis using the standard accumulative hypergeometric statistical test. Expression of DE genes on CD and HFD were normalized to z-scores by gene and colorized in circos heat map. The six most enriched pathways (PW) are shown.

All DE genes were submitted to STRING database for protein-protein interaction. The related protein interactions for DE genes enriched in the six PW were highlighted as links in the circos plot. **C** Gene set enrichment analysis (GSEA) showed significant up- and down-regulated pathways after HFD with false discovery rate (FDR) less than 0.05. All nominal $p$-values were less than 0.01. Normalized enrichment score (NES) reflects the degree to which a gene set is overrepresented in a ranked list of genes. **D** Illustration of transcriptomic changes related to metabolism after 2-day HFD. Raw sequencing data available in the GEO repository (GSE249790).

in vitro (Supplementary Fig. S6B). While mutation of the catalytic cystine (C603S) completely abolished SENP1 activity as expected, the C535S mutations maintained a 2-fold higher activity compared to the SENP1 WT and other mutants (Fig. 5B). SENP1 activity was also abolished in H533S or D550S mutants, the latter being partially rescued by concomitant C535S mutation (Fig. 5C). Consistent with proton shuffling from C603 via the nearby H533 towards D550 as a potential mechanism of activation (Supplementary Fig. S6C)[40], competing away suppressive effect of C535 interaction with H533 by excess histidine (1 mM) indeed prevents inactivation of SENP1 by 0.1 mM $H_2O_2$ (Fig. 5D).

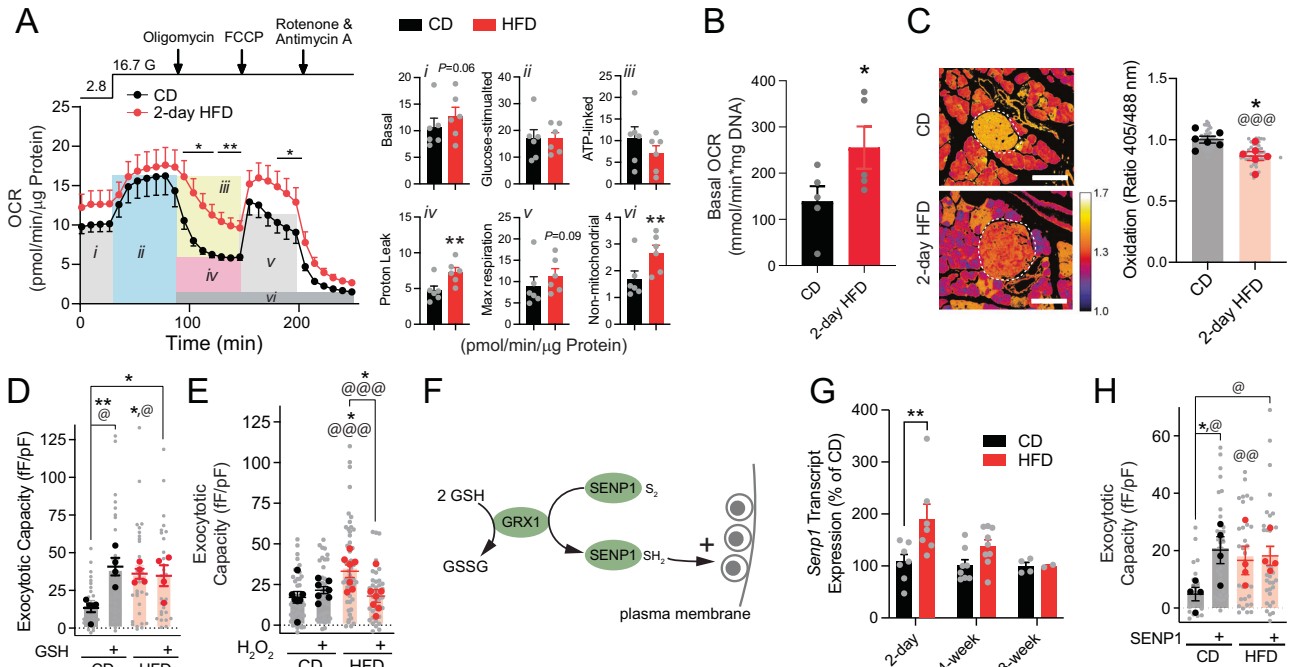

**Fig. 3 | Reductive cytosolic redox signaling via SENP1 contributes to enhanced exocytosis during 2-day HFD. A** Oxygen consumption rate (OCR) measured by Seahorse assay ($n = 6$ pairs of mice, $^*p = 0.017, 0.030$; $^{**}p = 0.0039$), with relevant respiration parameters calculated at right as shown by italicized numbers ($^{**}p = 0.0036, 0.0089$). **B** Basal OCR measured by Fluorescence Lifetime Micro Oxygen Monitoring at 2.8 mM glucose ($n = 4, 5$ mice; $^*p = 0.018$). **C** Representative image of pancreatic islets (white dashed circle) carrying cyto-roGFP2-Orp1 sensor (left). Scale bar = 100 μm. Redox ratio of individual islets ($n = 65, 41$ islets, $^{@@@}p = 1.3 \times 10^{-12}$, from 6 pairs of mice, $^*p = 0.041$). **D, E** Exocytosis with infusion of 10 μM GSH (panel **D**; $n = 27, 30, 31, 38$ cells, $^{@}p = 0.046, 0.017$; from 4 pairs of mice, $^*p = 0.026$, $^{**}p = 0.0073$) or 200 μM $H_2O_2$ (panel **E**; $n = 43, 39, 56, 39$ cells, $^{@@@}p = 7.1 \times 10^{-6}, 2.9 \times 10^{-4}$; from 7 pairs of mice, $^*p = 0.015, 0.022$) at 2.8 mM glucose after 2-day HFD. **F** Illustration of redox-control of insulin exocytosis. GRX1-

glutaredoxin 1; GSH – reduced glutathione; GSSG- glutathione; SENP1- sentrin-specific SUMO-protease 1. **G** *Senp1* expression by qRT-PCR in islets from CD or HFD fed mice ($n = 7, 7, 8, 9, 4, 2$ mice, $^{**}p = 0.0017$). **H** Exocytosis with infusion of 4 μg/mL catalytic SENP1 ($n = 21, 32, 31, 33$ cells, $^{@}p = 0.032, 0.026, ^{@@}p = 0.0044$; from 4 pairs of mice, $^*p = 0.18$) at 2.8 mM glucose. In panels **C, D, E, H** data are shown as individual cells (gray) or cells averaged by animal (dark). Data are mean ± SEM, compared with students paired t-test (two-sided; panels **A**–**C**) or RM two-way ANOVA with matching and Bonferroni post-test (panel **A**), or two-way ANOVA followed by Tukey post-test (panels **D, E, G, H**). Levels of significance are indicated for analysis with cells as replicates ('@') or with animals as replicates (*). @/*$P < 0.05$, @@/**$P < 0.01$, @@@/***$P < 0.001$ versus CD or as indicated. Source data are provided as a Source Data file.

In the original experiments shown in Fig. 5B, we believe differences in activity resulted from differential sensitivity to oxidation upon exposure to the air. Under conditions where SENP1 is fully activated (minimal oxidization and absence of divalent cations), the activities of SENP1 WT and SENP1 C535S were similar, and while the WT was strongly inhibited by $H_2O_2$, the C535S was not (Fig. 5E). Intracellular dialysis of SENP1 WT rescued the exocytotic defect (at 5 mM glucose) in β-cells from pSENP1-KO mice (Fig. 5F). While this rescue effect was reversed by co-dialysis of 10 μM $H_2O_2$, rescue of exocytosis in the pSENP1-KO by SENP1 C535S was resistant to $H_2O_2$ (Fig. 5F). Similar results were observed when SENP1 WT or SENP1 C535S were dialyzed into human β-cells (Supplementary Fig. S7A). The activity of SENP1 WT could be increased by GSH together with GRX1, which uses GSH as a cofactor to modulate protein thiols and disulfide bonds[41], while SENP1 C535S appeared maximally activated (Fig. 5G). Altogether these results suggest that C535 is required for SENP1 sensitivity to redox and its function in augmenting insulin exocytosis.

We wondered whether an additional mechanism contributes to suppression of baseline SENP1 activity. $Zn^{2+}$-cysteine is a critical mediator of redox-regulation and protease activity[42]. In silico analysis predicts that C603-C535-H533 is a $Zn^{2+}$-binding site in SENP1 (Supplementary Fig. S6D), and we found that $Zn^{2+}$, and to a lesser extent $Ni^{2+}$ (present during re-folding in some of our earlier experiments) inhibited the activity of SENP1 WT but not SENP1 C535S (Fig. 6A). The IC50 for SENP1 inhibition by $Zn^{2+}$ is right-shifted by nearly-30-fold in the C535S mutant, indicative of allosteric interaction of C535 with $Zn^{2+}$ to

inhibit SENP1 (Fig. 6B). This can be readily reversed by chelation with EDTA (Fig. 6C) and was not due to non-specific effect of $Zn^{2+}$ on His×6 tag from recombinant SENP1 (Supplementary Fig. S6E).

While $Zn^{2+}$ inhibits SENP1 WT with an IC50 (0.22 μM), similar to the well-established $Zn^{2+}$-inhibition of enzymes[43] including proteases such as the caspases (IC50 = 0.1–3.4 μM;[44,45]), this is above the free cytoplasmic $Zn^{2+}$ (sub-nM) in eukaryotic cells[46]. Compared to the in vitro $Zn^{2+}$ titration assay, regulation of $Zn^{2+}$ activity in a cellular environment is more complex, yielding compartmentalized $Zn^{2+}$ signaling events with defined spatiotemporal characteristics. We examined the potential involvement of metallothionein, a $Zn^{2+}$-binding protein participating in $Zn^{2+}$-storage and distribution to other proteins[47]. Metallothionein is abundantly present in the pancreas (average 264 μg/mL or 43.6 μM in human pancreas[48]). Because MT1X interfered with the binding of Hisx6 tagged SENP1 substrate (Hisx6-SUMO-mCherry) to Nickel-NTA agarose for the SENP1 assay, (see Methods), we developed an alternative assay that does not require Nickel-NTA agarose (Supplementary Fig. S6E). $Zn^{2+}$-bound metallothionein 1X (MT1X), a subtype present in β-cells, negatively regulates insulin secretion[49] and suppresses SENP1 activity similarly to free $Zn^{2+}$ (Fig. 6D), while $Zn^{2+}$-free MT1X; does not (Fig. 6E). $Zn^{2+}$ chelation with EDTA also reverses SENP1 inhibition by MT1X (Fig. 6F). In β-cells of the βSENP-WT mice, dialysis of 100 μM $ZnCl_2$ impaired exocytosis without decreasing $Ca^{2+}$ currents, to a level similar to that in the βSENP-KO (Fig. 6G; Supplementary Fig. S7B). While 100 μM $ZnCl_2$ did not decrease exocytosis further in the βSENP-KO, higher concentrations of

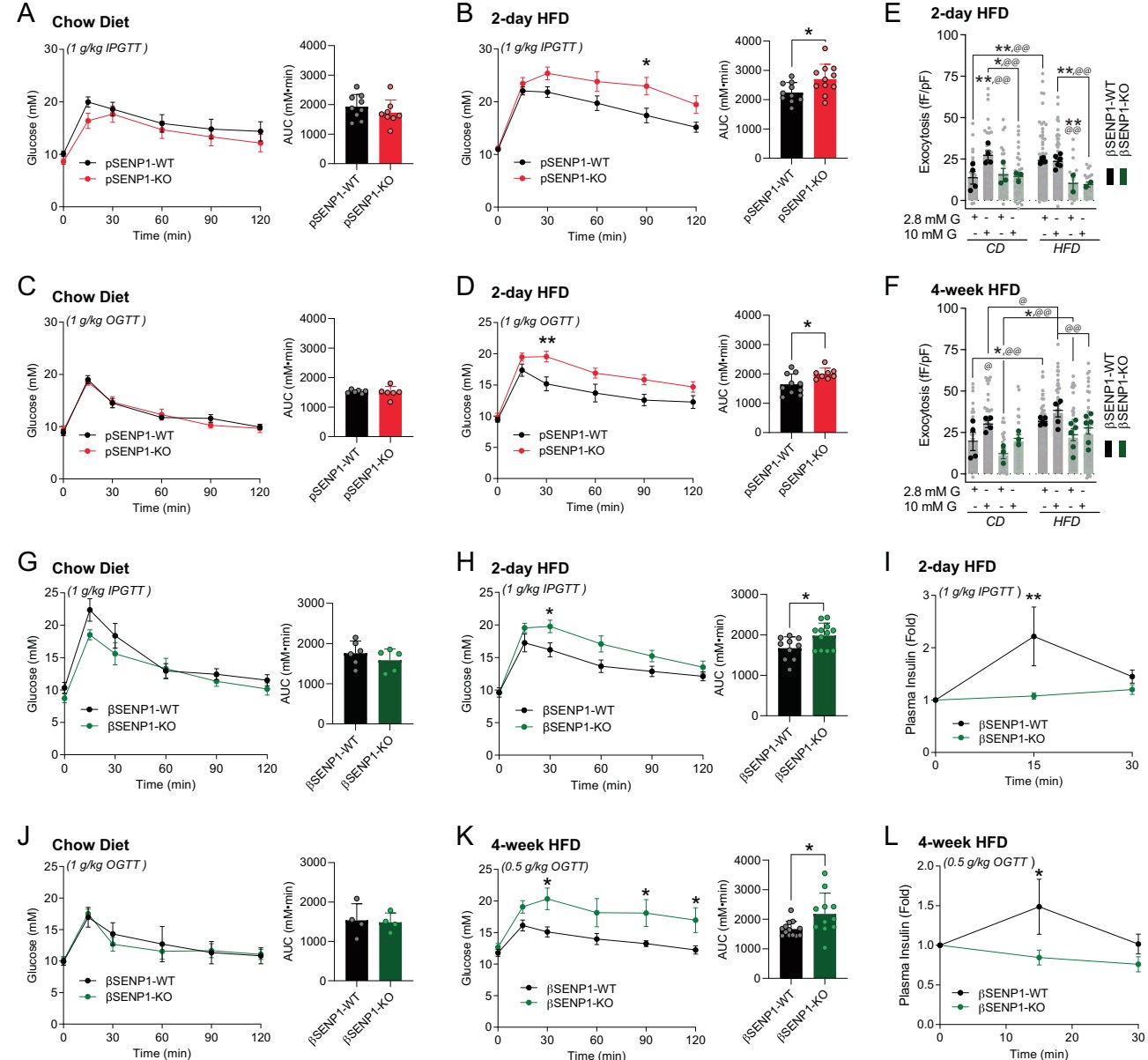

**Fig. 4 | βSENP1-KO mice develop glucose intolerance following 2-day HFD.**
**A**, **B** IPGTT of male pSENP1-KO and pSENP1-WT mice fed with CD (n = 9, 8 mice) and
2-HFD (n = 10, 11 mice, *p = 0.023, 0.027). **C**, **D** OGTT of male pSENP1-KO and
pSENP1-WT mice fed with CD (n = 6, 6 mice) and 2-HFD (n = 10, 8 mice, *p = 0.019,
**p = 0.0017). **E**, **F** Exocytosis from β-cells of βSENP1-WT and βSENP1-KO mice after
2-day (n = 34, 32, 24, 34, 56, 56, 18, 21 cells, @@p = 0.0017, 0.0018, 0.0032, 0.0030;
from 4, 3, 6, 2 mice, *p = 0.020, **p = 0.0029, 0.0036, 0.0092, 0.0097) and 4-week
HFD (n = 46, 45, 32, 31, 48, 56, 68, 63 cells, @p = 0.031, 0.021, @@p = 0.0011, 0.0017,
0.0044; from 4, 3, 4, 6 mice, *p = 0.027, 0.044). **G**, **H** IPGTT of male βSENP1-KO and
βSENP1-WT mice fed with CD (n = 6, 5 mice) and 2-day HFD (n = 10, 12 mice,

*p = 0.044, 0.025). **I** Plasma insulin levels during IPGTT after CD or 2-day HFD (n = 8,
8 mice, **p = 0.0049). **J**, **K** OGTT of male βSENP1-KO and βSENP1-WT mice fed with
CD (n = 4, 4 mice) and 4-week HFD (n = 13, 11 mice, *p = 0.021, 0.040, 0.050, 0.023).
**L** Plasma insulin levels during OGTT after CD or 4-week HFD (n = 8, 8 mice,
*p = 0.017). In panels **E** and **F** data are shown as individual cells (gray) or cells
averaged by animal (dark). Data are mean ± SEM, compared with student two-sided
unpaired Student's t test (AUCs) or two-way ANOVA followed either by Bonferroni
(two-sided; panels **A**–**D**, **G**–**L**) or Tukey (two-sided; panels **E**, **F**) post-test. Source
data are provided as a Source Data file.

ZnCl₂ (1 mM) dramatically decreased exocytosis in both groups pos-
sibly by inhibiting Ca²⁺ currents directly[50] (Fig. 6G, Supplementary Fig.
S7B). Finally, in the presence of Zn²⁺ SENP1 was more sensitive to
inactivation by H₂O₂ (Fig. 6H), and more robustly activated by GSH and
GRX1 (5.3-fold) while the C535S mutant remained more active (Fig. 6I).

## Discussion
β-cell compensation and decompensation is a determinant of early
T2D progression[51], yet the underlying mechanisms remain unknown.
Here we demonstrate that a reduced cytosol acting via SENP1 facil-
itates an upregulation of β-cell function to maintain glucose tolerance

very early after high fat feeding. Redox-sensing by SENP1 relies on C535
and is tuned by an interaction with Zn²⁺, possibly delivered via MT1X
(Supplementary Fig. S7C). Notably, the enhanced depolarization-
induced exocytosis observed at low glucose soon after 2-day HFD is
recapitulated by intracellular dialysis of reducing molecules or active
SENP1, which exerts a similar priming effect[21], and is lost upon β-cell
knockout of SENP1. This is not expected to increase basal insulin
secretion in the absence of an increased intracellular Ca²⁺ concentra-
tion. Instead, an increased insulin granule priming at low glucose will
increase the size of the releasable pool of insulin granules on which
subsequent glucose-dependent Ca²⁺ responses can act, resulting in

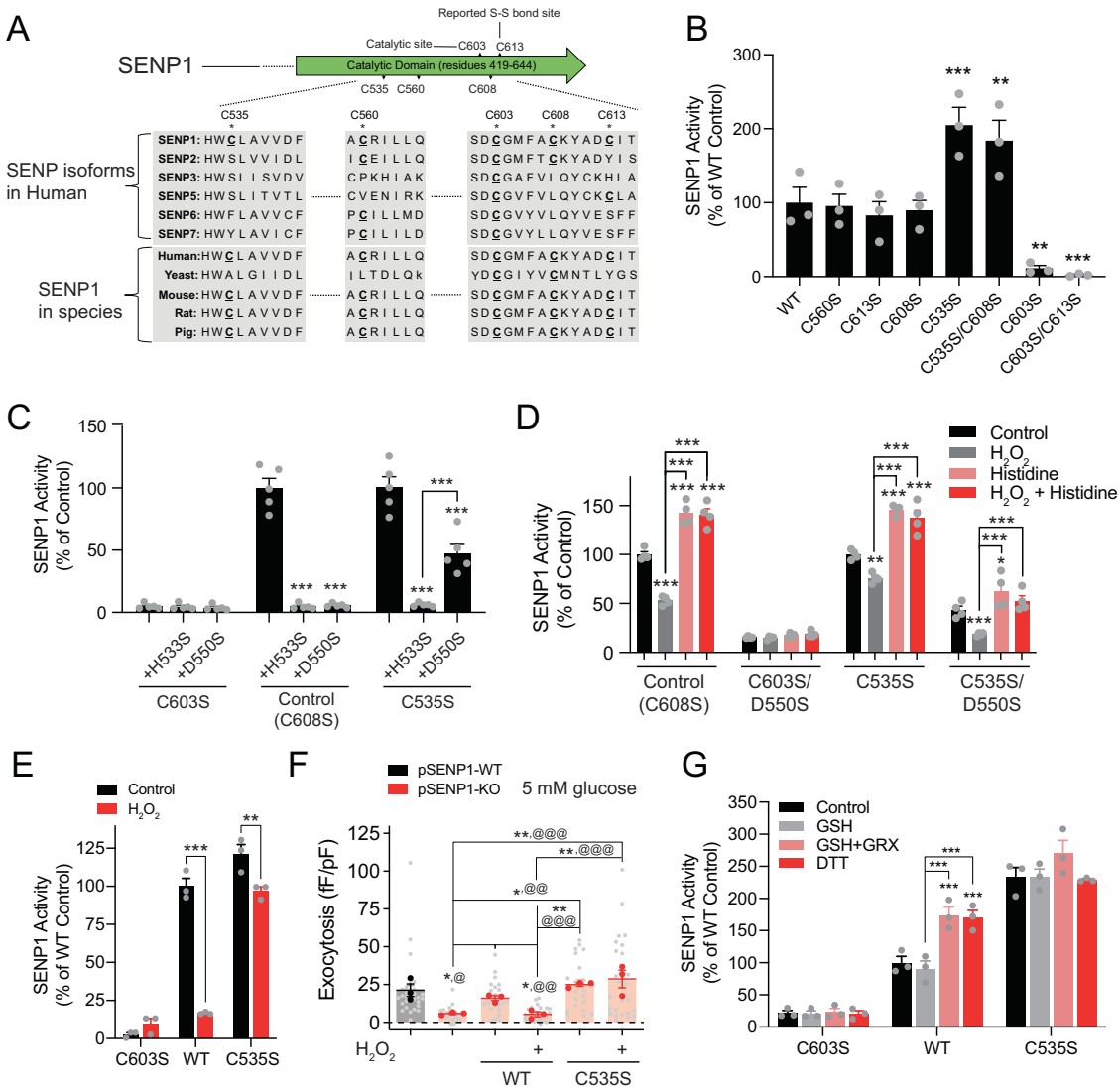

**Fig. 5 | Redox regulation of SENP1 activity requires C535. A** Multiple amino acid comparison among different isoforms of SENP across species. **B** SENP1 activity after indicated cysteine-to-serine substitution ($n = 3$ experiments, ** $p = 0.0034$, $0.0020$, *** $p = 0.00040$, $0.00079$). **C** SENP1 activity after cysteine-to-serine mutation and subsequent mutation of histidine 533 or aspartic acid 550 to serine ($n = 5$ experiments, *** $p = <1 \times 10^{-15}$, $<1 \times 10^{-15}$, $<1 \times 10^{-15}$, $6.1 \times 10^{-7}$). Control here was C608S that does not affect activity, but used to balance thiol groups. **D** SENP1 activity of control (C608S) SENP1 compared with serine mutants in the presence of $H_2O_2$ (100 μM), excess histidine (1 mM), or both ($n = 4$ experiments, Control: *** $p = 2.6 \times 10^{-9}$, $<1 \times 10^{-15}$, $<1 \times 10^{-15}$, $3.8 \times 10^{-8}$, $9.8 \times 10^{-8}$; C535S: ** $p = 0.0013$, *** $p = 4.6 \times 10^{-9}$, $<1 \times 10^{-15}$, $<1 \times 10^{-15}$ $5.9 \times 10^{-7}$; C535S/D550S: * $p = 0.012$, *** $p = 8.3 \times 10^{-4}$, $1.3 \times 10^{-8}$, $4.9 \times 10^{-6}$). **E** Activity of SENP1 C603S, WT, and C535S upon full activation by DTT (10 mM), and subsequent inhibition by 5 mM $H_2O_2$ ($n = 3$

experiments, ** $p = 0.0024$, *** $p = 0.9.6 \times 10^{-9}$). **F** In β-cells from pSENP1-KO (red) mice the effect of 4 μg/mL glutathione-S-transferase (GST) peptide (Control), SENP1 WT, or C535S were infused with/without 10 μM $H_2O_2$ on exocytosis at 5 mM glucose compared with pSENP-WT β-cells (black) ($n = 30$, 21, 24, 26, 21, 29 cells, @ $p = 0.015$, @@ $p = 0.0062$, $0.0025$, @@@ $p = 5.4 \times 10^{-5}$, $9.6 \times 10^{-4}$, $1.1 \times 10^{-5}$; from 3, 3 mice, * $p = 0.040$, $0.033$, $0.011$, ** $p = 0.0031$, $0.0089$, $0.0026$). **G** Activity of SENP1 C603S, WT, and C535S in the presence of GSH (0.1 mM) alone or with GRX1 (10 μg/ml). DTT (10 mM) was used to fully activate the enzymes ($n = 3$ experiments, *** $p = 3.3 \times 10^{-4}$, $5.8 \times 10^{-4}$, $7.5 \times 10^{-5}$, $1.3 \times 10^{-4}$). In panel F data are shown as individual cells (gray) or cells averaged by animal (dark). Data are mean ± SEM, compared with RM one-way ANOVA followed by Tukey post-test (panels **B**, **F**) or two-way ANOVA followed by Tukey (panels **C**, **D**, **G**) or Bonferroni (panel **E**) post-test within groups. Source data are provided as a Source Data file.

increased glucose-stimulated insulin secretion. This is consistent with the enhanced efficacy of $Ca^{2+}$ in 1-week HFD-fed mice[5] and pre-diabetic *db/db* mice[15].

Redox signals are important regulators of insulin secretion and functional compensation[12,17], although there is debate as to whether reactive oxygen species[52] or reducing signals[53] augment insulin secretion. Possibly both are true, depending on timing, localization, and mechanisms involved, such as the modulation of excitability by reactive oxygen species via K+ channels[54] or by the direct regulation of membrane fusion by reducing signals[35]. Reducing signals may also be 'permissive' through their role in maintaining or setting a robust pool of releasable granules before subsequent glucose stimulation[52]. We

observed a remodeling of metabolic pathways in RNAseq data and increased proton leak and basal $O_2$ consumption after 2-day HFD. Coincident with this, we observed a more reduced cytosol in the β-cell. Although the underlying mechanism remains unclear, down-regulation of cholesterol biosynthesis as seen in the RNAseq data may contribute to an increase in cytosolic NADPH[33]. Increased mitochondrial proton-leak may also contribute to higher insulin secretion[55], and increased sub-maximal glucose-fueled mitochondrial TCA cycle flux may generate higher cytosolic reducing equivalents[56]. We and others have shown that NADPH, produced by either the mitochondrial export of reducing equivalents or the pentose phosphate pathway[57,58], augments $Ca^{2+}$-triggered insulin exocytosis[17,35,36]. One caveat is that the

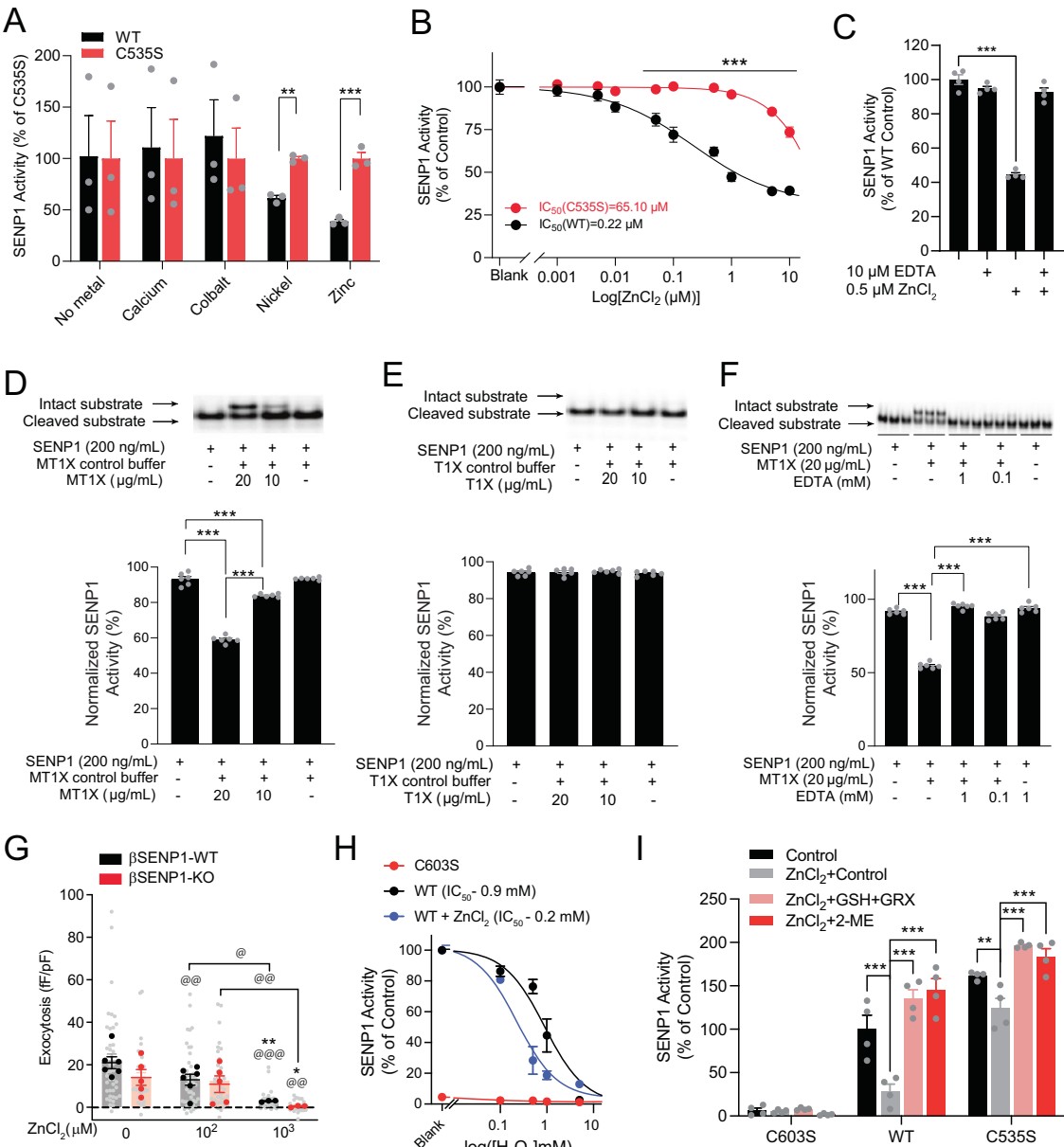

**Fig. 6 | Zn²⁺ tunes SENP1 redox sensitivity and SENP1-dependent β-cell exocytosis. A** Activity of SENP1 WT and C535S in the presence or absence of Ca²⁺, Co²⁺, Ni²⁺ and Zn²⁺ ($n = 3$ experiments, **$p = 0.0071$, ***$p = 0.00029$). Data were normalized to C535S activity, as this is more resistant to oxidation. **B** Dose-response curve of SENP1 inhibition by ZnCl₂ ($n = 3$ experiments, ***$p = 1.9 \times 10^{-14}$). **C** SENP1 inhibition by ZnCl₂ could be reversed with the chelating reagent EDTA ($n = 4$ experiments, ***$p = 1.6 \times 10^{-8}$). **D**–**F** SENP1 activity, measured using a native-PAGE assay, inhibited by Zn²⁺-carrying (MT1X, panel **D**, ***$p = <1 \times 10^{-15}$, $1.5 \times 10^{-6}$, $1.5 \times 10^{-12}$) but not Zn²⁺-depleted (T1X, panel **E**) metallothionine or by Zn²⁺-carrying MT1X in the presence of EDTA (panel **F**, ***$p =$ all are $1.9 \times 10^{-14}$) ($n = 6$ experiments). **G** Effect of Zn²⁺ on exocytosis from β-cells of βSENP1-WT and -KO mice at 5 mM glucose ($n = 46, 35, 46, 39, 23, 23$ cells, @$p = 0.023$, @@$p = 0.0035, 0.0041, 0.0012$, @@@$p = 7.4 \times 10^{-7}$; from 6, 6, 3

pairs of mice, *$p = 0.030$, **$p = 0.030$). **H** Concentration-response curve of SENP1 activity, with enzyme prepared with or without ZnCl₂ during refolding, to inhibition by H₂O₂. Activity of SENP1 C603S is shown for comparison ($n = 6$ experiments). **I** Activity of SENP1 C603S, WT, and C535S in the presence of 1 μM Zn²⁺ and subsequent activation with 5 mM GSH and 10 μg/ml GRX1. 2-mercaptoethanol (2-ME, 1 mM) was used to fully activate enzymes ($n = 4$ experiments, *$p = 0.014$, ***$p = 3.5 \times 10^{-6}$, $5.1 \times 10^{-10}$, $5.1 \times 10^{-11}$, $2.6 \times 10^{-6}$, $6.5 \times 10^{-5}$). In panel G data are shown as individual cells (gray) or cells averaged by animal (dark). Data are mean ± SEM, compared with RM one-way ANOVA followed by Tukey post-test (panels **C**–**F**) or RM two-way ANOVA followed by Bonferroni post-test within groups (panels **A**, **I**) or across conditions (panels **B**, **G**). Source data are provided as a Source Data file.

Cyto-roGFP2-Orp1 redox-sensor reports the steady state balance between oxidizing and reducing signals[20,59]. The reduced redox could result from either decreased H₂O₂ production or increased reducing equivalents. Adaptive glucose metabolism may also enhance de novo GSH biosynthesis independent of NADPH generation[60]. Therefore, increased de novo GSH biosynthesis could contribute to a compensatory increase in insulin secretion. Nonetheless, a reduced cytosolic redox state appears critical for the increased β-cell exocytosis seen

after 2-day HFD since it could be reversed by H₂O₂ and replicated by GSH or NADPH.

The SUMO-protease activity of SENP1 is required for its ability to augment insulin exocytosis (Fig. S7A), consistent with our previous report that SENP1 can rescue exocytosis suppressed by SUMO1 or the SUMO-ligase Ubc9, and deSUMOylates synaptotagmin VII[21]. Compared to other isoforms, SENP1 is a redox-sensitive protease with an unique interaction between a free thiol on the catalytic cysteine (C603) and

other cysteines[18]. C613 is reported to be a 'positive' redox sensor by forming an intermolecular disulfide bond with the C603 and protect SENP1 from irreversible oxidation[18]. We find however that mutation of C613 had little effect on SENP1 activity, but rather that C535 appears to restrain SUMO-protease activity. The C535S mutant had 2-fold higher activity and was resistant to inactivation by $H_2O_2$, indicating that C535 is a 'negative' modulator of redox sensing. Indeed, the atomic distance between thiols of C535 and C603 prevents formation of an intra-molecular disulfide bond, consistent with the absence of disulfide bonds in available SENP1 crystal structures[39]. C535 also does not control enzyme activity via the formation of intermolecular SENP1 disulfide bonds[18]. Thus, it seems most likely that C535 impacts SENP1 redox-sensing via a proton transfer pathway, and deprotonation of the thiol group on C603 to activate SENP1 (Supplementary Fig. S6C). Specifically, proton transfer from the thiol group of C603 towards the carboxyl group of D550 via H533 results in the conversion of a thiol on C603 to an active thiolate in SENP1[40]. C535 may donate a proton to either H533 or D550 to competitively restrain proton transfer from C603, leading to inhibition of SENP1, under oxidizing conditions.

SENP1 couples reducing power to insulin secretion[17,21,22,61]. Interestingly, we find that $Zn^{2+}$ allosterically inhibits SENP1 in a manner that is dependent upon C535. Mechanistically, the predicted $Zn^{2+}$-binding site includes C603, H533 and C535, which is one of the most ubiquitous $Zn^{2+}$ binding motifs[62]. A coordinate bond between $Zn^{2+}$ and these residues likely further restricts proton transfer. $Zn^{2+}$ coordination is expected to lower the pKa of a thiol by two orders of magnitude, facilitating the formation of a thiolate[63]. Indeed, thiol-binding $Zn^{2+}$ readily dissociates in the presence of oxidant, leaving a highly reactive thiolate for an oxidant[64]. Therefore, interaction between C535 and $Zn^{2+}$ sensitizes SENP1 to oxidative inhibition and allows robust reactivation by reducing signals, not dissimilar to the reported interaction between reactive oxygen species and $Zn^{2+}$ in the regulation of protein tyrosine phosphatases[65]. This is consistent with the recent demonstration of $Zn^{2+}$-lactate binding at C535 of SENP1[66] which, together with the recent finding that lactate is higher than expected in some human β-cells[67], suggests an additional possible mechanism for tuning β-cell exocytosis.

$Zn^{2+}$ inhibits exocytosis via SENP1 with a calculated free $Zn^{2+}$ at ~0.08 μM. However, cytosolic free $Zn^{2+}$ levels are not perturbed by glucose stimulation[68]. $Zn^{2+}$ binding of SENP1 may occur at the time of protein folding in the ER, or via direct interaction with $Zn^{2+}$-binding/buffering proteins such as MT1X which have been shown to control islet $Zn^{2+}$ fluctuations[69] and inhibit glucose-stimulated insulin secretion via a so-far unidentified interaction with the exocytotic machinery[49]. While free $Zn^{2+}$ is in the low-nM-range in the β-cell cytosol[68], the abundant $Zn^{2+}$-binding MT1X likely transfers $Zn^{2+}$ to SENP1, tuning its redox-sensitivity and activity. During β-cell compensation, MT1X was downregulated to increase insulin secretion[49]. The combined effect of less $Zn^{2+}$ transferred to SENP1, upregulation of *Senp1* expression, and a reduced redox state would be expected to result in a more active SENP1 and augmentation of β-cell exocytosis.

Knockout of islet and β-cell SENP1, in two separate Cre-driver models, led to IP glucose intolerance after 2-day HFD associated with a loss of insulin response. This was more obvious in males, which are known to be more sensitive to HFD[37] and consistent with differences we observed in human islets, where insulin secretion and exocytosis appeared most impacted by BMI in younger males. However, oral glucose tolerance was not different following 2-day HFD in most cases and in contrast with the 4-week HFD where the SENP1-KO showed selectively impaired oral, not IP, glucose tolerance. This discrepancy highlights the adaptive changes that occur during different stages of high fat feeding[70]. The incretin response is paramount in limiting oral glucose intolerance after long-term HFD[24,71], where incretin release from the gut[24] and β-cell incretin receptors[70] are upregulated. β-cell SENP1 is required for these compensational changes to maintain incretin-stimulated insulin secretion and oral glucose tolerance[24,72]. However, the β-cell GLP-1 receptor is not necessary for oral glucose tolerance under normal conditions[73], similar to the role of SENP1 for oral glucose tolerance under the same conditions[24] or after 2-day HFD. Additionally, the extra-islet action of GLP-1 could mask the effect of β-cell SENP1, such that OGTT is similar in βSENP1-KO and -WT mice[73]. Finally, we note that the less cell-type-selective pSENP1-KO mice, which have impaired GIP-stimulated glucagon secretion[24], did show impaired oral glucose tolerance after 2-day HFD, indicating the potential importance of α-to-β cell communication under this condition and as previously reported[71].

Thus, we show that insulin secretion is upregulated from islets from human donors with increased BMI, at early/mid-age, and from mouse islets shortly after high fat feeding. This is not due to an increased $Ca^{2+}$ response in β-cells, but to an increase in the availability of releasable insulin granules that is promoted at low glucose (i.e., prior to the glucose-stimulated secretion of these granules). This requires a reducing environment and activity of the SUMO-protease SENP1 via a sensing mechanism involving the $Zn^{2+}$-dependent redox control of enzyme activity by C535 near the enzyme catalytic site. In mice this enhanced insulin secretory capacity acts to maintain glucose tolerance in the early stages of high fat feeding.

## Methods

All human islet studies were approved by the Human Research Ethics Board (Pro00013094; Pro00001754) at the University of Alberta and all families of organ donors provided written informed consent. All studies with mice were approved by the Animal Policy and Welfare Committee (AUP00000291) at the University of Alberta.

### Islet isolation and cell culture

Human islets were isolated at the Alberta Diabetes Institute IsletCore (www.isletcore.ca) and cultured overnight in DMEM (11885, Gibco) supplemented with L-glutamine, 110 mg/l sodium pyruvate, 10% FBS (12483, Gibco), and 100 U/ml penicillin/streptomycin (15140, Gibco)[74]. A total of 265 non-diabetic (ND) donors and 34 T2D donors (HbA1c >6.5) (Supplementary Data 1) were examined. Separation of donors by BMI and age was based on clinical diagnostic criteria for overweight, and previous studies by us[74] and others[75] demonstrating declines in human islet insulin secretion between the ages of 40 and 50.

C57BL/6NCrl mice (Charles River Laboratories) of both sexes were fed standard chow diet (5L0D*, PicoLab Laboratory Rodent Diet) or 60% high-fat diet (S3282, Bio-Serv) at 12 weeks of age for 2-days or 4-weeks. Pdx1-Cre mice (B6.FVB-Tg (Pdx1-Cre)6 Tuv/J, Jackson Lab, 014647) on a C57BL/6N background and Ins1-Cre mice on a mixed C57BL/6J and SV129 background[76] were crossed with *Senp1*-floxed mice on a C57BL/6J background to generate gut/pancreas specific (Pdx1-Cre$^+$;SENP1$^{fl/fl}$ - pSENP1-KO) and β-cell-specific knockouts (Ins1-Cre$^+$;SENP1$^{fl/fl}$ - βSENP1-KO)[17,77]. Control littermates Pdx1-Cre$^+$;SENP1$^{+/+}$ mice (pSENP1-WT) and Ins1-Cre$^+$;SENP1$^{+/+}$ mice (βSENP1-WT) were used for experiments. All mice were 12-16 weeks of age at the time of experiments and housed under a 14/10 light-dark cycle with *ad libitum* access to food. Genotypes and SENP1 expression were confirmed as previously[24]. Following euthanasia by $CO_2$ inhalation, mouse islets were isolated by collagenase digestion and purified by histopaque density gradient centrifugation before hand-picking[24]. All mouse islets or single cells were cultured in RPMI-1640 (11875, Gibco) with 11.1 mM glucose, 10 % FBS (12483, Gibco), 100 units/mL penicillin, and 100 μg/mL streptomycin (15140, Gibco).

### Insulin secretion

After overnight culture, static insulin secretion was measured as previously[74] at the glucose concentrations indicated. Diazoxide and KCl concentration were 100 μM and 30 mM respectively. For dynamic

insulin responses 25 islets were pre-perifused for 30 min at 2.8 mM glucose followed by 16.7 mM glucose and KCl (30 mM) at a sample collection rate of 100 μl/min every 2–5 min[24]. Insulin content was extracted with acid/ethanol. The samples were stored at −20 °C and assayed by using Insulin Detection Kit (STELLUX® Chemi Rodent Insulin ELISA kit; STELLUX® Chemi Human Insulin ELISA kit, Alpco).

## Exocytosis measurements

Human and mouse islets were dispersed to single cells in $Ca^{2+}$-free buffer and cultured in 5.5 mM glucose DMEM and 11 mM glucose RPMI, respectively as above. After overnight culture, dispersed cells were preincubated at 1 mM (human) or 2.8 mM (mouse) glucose for 1 h and patched in bath solution containing (in mM): 118 NaCl, 5.6 KCl, 20 TEA, 1.2 $MgCl_2$, 2.6 $CaCl_2$, 5 HEPES at different glucose conditions (1, 2.8, 5, 10 mM) with a pH of 7.4 (adjusted by NaOH) at 32-35 °C. Whole-cell capacitance was recorded with the sine+DC lock-in function of an EPC10 amplifier and PatchMaster software (HEKA Electronics). Exocytotic responses and inward $Ca^{2+}$ currents were measured 1-2 min after obtaining the whole-cell configuration in response to ten 500 ms depolarizations to 0 mV from a holding potential of −70 mV. Changes in capacitance and integrated $Ca^{2+}$ charge entry were normalized to cell size (fF/pF and pC/pF, respectively). The intracellular solution contained (in mM): 125 Cs-Glutamate, 10 CsCl, 10 NaCl, 1 $MgCl_2$, 5 HEPES, 0.05 EGTA, 3 MgATP and 0.1 cAMP with pH = 7.15 (pH adjusted with CsOH).

In the intracellular dialysis experiments, measurements were 2-3 min after obtaining the whole-cell configuration. Our previous work showed this is sufficient for rapid dialysis of both small molecules and larger peptides such as SENP1[21] and given the large volume of the pipette solution with respect to the β-cell cytosol we assume a homogenous distribution of infused molecules. Compounds or recombinant enzymes were added to pipette solution as indicated. For mouse β cells, these were 10 μM $H_2O_2$ (Fisher Scientific, H325-500), 10 μM reduced glutathione (GSH, Sigma), 4 μg/ml cSENP1 (Enzo Life Technologies) or glutathione-S-transferase (GST) (Enzo Life Technologies) as a control protein. For human β-cells, these were 4 μg/ml GST, wild type SENP1, C535S and C603 mutant recombinant proteins (see below) pretreated with or without 200 μM $H_2O_2$ for 10 min. Human β-cells were identified by immunostaining for insulin (Dako anti-insulin, #IR002 at 1:5 dilution, with Invitrogen AF488 secondary, #A11073 at 1:200 dilution). Mouse β-cells were identified by size (>4 pF) and a $Na^+$ channel half-maximal inactivation at around −90 mV[78].

## Glucose homeostasis

Oral glucose tolerance test (OGTT) and intraperitoneal tolerance test (IPGTT) were performed as previously described[24]. For OGTT and IPGTT on chow diet or 2-day HFD, glucose concentration was 1 g/kg dextrose. After 4-week HFD mice became glucose intolerant and 1 g/kg dextrose produced blood glucose values above the maximum of the glucometer, and we therefore used 0.5 g/kg dextrose in these experiments. Tail blood was collected at indicated times for insulin (STELLUX® Chemi Rodent Insulin ELISA kit, Alpco) and glucose measurement. ITT was performed with a concentration of 1 U/kg Humulin R (Eli Lilly).

## Intracellular $Ca^{2+}$ imaging

Mouse islet cells were cultured on 35 mm glass bottom dishes with 10 mm micro-well (Cellvis) overnight in 11 mM glucose RPMI media (11875, Thermo Fisher). Cells were pre-incubated with Fura-2AM (1 μM) for 15 min and perifused with KRB solution at indicated glucose level and imaged at 0.5 Hz. Fluorescence signal was excited at 340/380 nm (intensity ratio 20:8) and detected at an emission light of 510 nm using Life Acquisition software (Till Photonics) on an inverted microscope (Zeiss Axioobserver, Carl Zeiss Canada Ltd.) equipped with a rapid-switching light source (Oligochrome; Till Photonics, Grafelfing, Germany). β-cells were marked and identified by immunostaining and fluorescence ratios were calculated using ImageJ (NIH).

## Fluorescence activated cell sorting (FACS) of β-cells and mRNA sequencing

Islet α- and β-cells were identified by ZIGIR vs Ex4-Cy5 through FACS analysis as above[79]. Total RNA was isolated from sorted β-cells using TRIzol reagent (Invitrogen) according to the manufacturer's protocol and cDNA libraries were prepared using Illumina's TruSeq Stranded mRNA Library Prep kit. The cDNA libraries were validated by TapeStation DNA 1000 High Sensitivity assay (Agilent) and quantified by Qubit dsDNA High Sensitivity assay (Invitrogen). High quality libraries were submitted to the UT Southwestern Next Generation Sequencing Core facility and RNA sequencing was performed using Illumina's NextSeq 500 High Output instrument. Raw reads were processed to transcripts per million (TPM) and differentially expressed (DE) genes were initially determined by using the edgeR package. Genes with $P$-value <0.1 (not corrected for multiple testing), coefficient of variation (CV) of less than 1, same change of direction (either positive or negative) and specific gene counts range (mean gene counts in at least one experimental group should be higher than 5 RPKM) were considered differentially expressed between CD and HFD. All differentially expressed genes are used for downstream analyses and listed in Supplementary Data 2.

Differentially expressed gene function and enriched pathways were identified with Metascape (http://metascape.org/). KEGG Pathway, GO Biological Processes, Reactome Gene Sets, CORUM, TRRUST and PaGenBase were included in pathway enrichment analysis with a $p$-value <0.01. Terms with an enrichment factor >1.5 and a similarity of >0.3 are grouped into a cluster represented by the most significant term[80]. To examine protein-protein interaction (PPI), STRING database (https://string-db.org/) was used and only PPIs with interaction score higher than 0.70 were retrieved and linked in the circus plot by using R-studio (circlize package)[31,81]. To run an unbiased gene set enrichment analysis (GSEA)[82], all Genes with CV of less than 1, same change of direction and specific gene counts range (at least one experimental group should be higher than 5 RPKM) were submitted and listed in Supplementary Data 3. H.all.v7.4.symbols.gmt[Hallmarks] gene dataset was used as reference data set. Enriched gene sets were considered significant with gene size over 50, false discovery rate less than 0.05, nominal $p$-value less than 0.01.

## Redox measurement

Measurements of the intracellular $H_2O_2$ levels were made using a redox histology approach, as previously described[34,83,84]. Briefly, batches of 30 islets were collected and immersed in 50 mM N-ethyl-maleimide (NEM) dissolved in PBS for 20 min for sensor chemical fixation. The NEM was then removed, and islets fixed in 4% paraformaldehyde for 30 min. After paraformaldehyde removal, the islets were incubated in 100-μL HepatoQuick (Roth, Karlsruhe, Germany) mixed with human citrate plasma (1:2 v/v) and 1 % $CaCl_2$ for 1 h at 37 °C. Clots were then incubated in 95 % ethanol at 4 °C overnight and subsequently dehydrated in ethanol prior to paraffin embedding. The paraffin embedded clots were cut in 3 μm thick sections with a semi-automated rotary microtome (Leica Biosystems, Wetzlar, Germany) and placed on silanized glass slides. Images of all the islets in each slide were obtained by the Axio Observer 7 fluorescence microscope (Zeiss, Oberkochen, Germany) with a 20× objective using excitation 405 nm and 488 nm, emission 500-530 nm. The images were analyzed with ImageJ Fiji Software, and the ratio (405/488 nm) was used to compare different groups.

## Oxygen consumption

Agilent's seahorse XFE24 Analyzer with the islet capture microplates was used for oxygen consumption measurements[85]. Briefly, 70 islets per well were assayed. Islets were sequentially treated with 2.8 mM glucose, 16.7 mM glucose, 5 μM oligomycin, 5 μM FCCP and 5 μM rotenone/antimycin A. Each experiment was run with a 3-min mix, 2-min wait and 3-min measurement period for all data points. Each biological replicate had three technical replicates and oxygen consumption rate was normalized to islet protein.

For oxygen consumption measured with the Fluorescence Lifetime Micro Oxygen Monitoring System (FOL/C3T175P, Instech Laboratories Inc.) around 200 islets incubated overnight were washed 3x with serum-free RPMI media at 2.8 mM glucose, added to the chamber for data collection for 20 min, and then collected for normalization of oxygen consumption to DNA content measured using a Quant-iT™ PicoGreen™ dsDNA Assay Kits (Invitrogen).

## Preparation of recombinant wild-type and mutant SENP1

Human SENP1 catalytic domain cDNA (NM_001267594/NP_00125 4523.1, 419-644 residues) was cloned in the downstream of a His×6 coding sequence, between Xba-I and BamH-I sites of the pT7JLH plasmid to add a His×6 N-terminal tag to SENP1. Primers are listed in Supplementary Data 5. Site directed mutagenesis by primer extension was used to make SENP1 mutants, C535S, C560S, C603S, C608S, C613S, C535S/C608S, C603S/C613S, C535S/H533, C535S/D550, C603S/H533, C603S/D550, C608S/H533 and C608S/D550. In brief, two sets of overwrapping mutagenesis primers were used to amplify SENP1 catalytic domain using Phusion DNA polymerase. The SENP1 cDNA was cloned between Xba-I and BamH-I sites using Gibson assembly.

To produce wild type SENP1 without His×6 tag, human SENP1 catalytic domain cDNA was also cloned into immediately downstream of T7 promoter, between Nde-I and BamH-I of the pT7JLH plasmid, to avoid addition of a His×6 tag coding sequence.

For expression, Rosetta strain of *E. coli* (Novagen) was transformed with each expression vector and grown in 50 ml LB broth containing 100 μg/ml ampicillin at 37 °C with shaking until A600 reached between 0.6 and 0.8. Each SENP1 protein was overexpressed by addition of 1 mM IPTG to the culture. After 2 h induction, the *E. coli* pellet was harvested by centrifugation at 10,000 × *g* for 10 min. Inclusion bodies of SENP1 proteins were prepared from the pellet[86].

Each SENP1 protein with a His×6 tag was solubilized in 6 M guanidine HCl, 20 mM Tris, 0.1% Tween20, 1 mM 2-mercaptoethanol, pH = 8, and captured with a HisPur Nickel-NTA resin column (1.5 × 3 cm). The column was washed with 4 M guanidine HCl, 20 mM Tris, 0.1% Tween20, 1 mM 2-mercaptoethanol, pH = 8 and then SENP1 protein was eluted with 4 M guanidine HCl, 50 mM sodium acetate, 0.05% Tween20, pH = 4. To remove Ni²⁺ leached out from nickel-NTA resin, the concentration of each SENP1 protein was adjusted to 4 mg/ml with 4 M guanidine HCl, 20 mM Tris, 0.1% Tween20, 1 mM DTT, 50 mM EDTA and 0.2 M Tris, pH = 8. Ten volumes of water were added to dilute the 4 M guanidine HCl to 0.4 M, which causes the precipitation of SENP1 protein. The precipitated SENP1 protein without nickel was recovered by centrifugation at 8000 × *g* for 20 min.

The inclusion bodies of wild type SENP1 without a His×6 tag, was further washed twice with 1 M guanidine HCl, 2% Triton X-100, 0.1 M Tris, pH = 8. The inclusion bodies were dissolved in 4 M guanidine HCl, 20 mM Tris, 0.1 % Tween20, 1 mM DTT, 50 mM EDTA and 0.2 M Tris, pH = 8. The SENP1 without a His×6 tag was precipitated by addition of ten volumes of water. The SENP1 without a His×6 tag was recovered by centrifugation at 8000 × *g* for 20 min.

The precipitated SENP1 was dissolved in 6 M guanidine HCl, 20 mM Tris, 0.2 M Arginine, 1 mM DTT, pH = 7.2 and the absorbance at 280 nm was measured to estimate protein concentration. SENP1 proteins were stored at −80 °C until in vitro refolding. Refolding of SENP1

protein was carried out in vitro by gradual decrease of guanidine HCl concentration with dialysis[87] at 4 °C. The concentration of SENP1 protein was adjusted to 1 mg/ml with 6 M guanidine HCl, 20 mM Tris, 0.1% Tween20, 0.2 M arginine, pH = 7.2, and for each refolding 4-5 mg of SENP1 was used. The cysteines of SENP1 proteins were reduced to have sulfhydryl side chains with 10 mM DTT for 1 h at room temperature and dialyzed against 0.5 M guanidine HCl, 0.2 M arginine, 0.1 M Tris, 2 mM cysteine, 0.2 mM cystine, 0.05% Tween20, pH = 7.2 for 24 h, and then against the same solution without guanidine HCl for 24 h. SENP1 proteins were refolded with/without 0.1 mM ZnCl₂, and then ZnCl₂ was removed by dialysis. Alternatively, SENP1 and mutants were prepared without ZnCl₂ in the refolding buffer then incubated with 1 μM ZnCl₂ for 1 h to suppress SENP1 activities prior to reactivation by 0.1 mM GSH + 10 μg/ml GRX1. SENP1 proteins were finally dialyzed against SENP1 assay buffer, 20 mM Tris, 0.1 M sodium chloride, 0.05 % Tween20, pH = 7.2 for 24 h. SENP1 proteins, used for the intracellular dialysis experiments, were prepared without 0.05% Tween 20. The final dialysis buffer was changed to fresh buffer twice. The precipitated SENP1 proteins were removed by centrifugation at 8000 × *g* for 10 min. Absorbance at 280 nm of the supernatant were measured to estimate the concentration of SENP1 proteins.

For the experiments to study the effect of divalent transient metals on SENP1 activity, 0.1 mM of CaCl₂, CoCl₂, NiCl₂ and ZnCl₂ was added in the refolding solution, and then removed by extensive dialysis against SENP1 assay buffer, 20 mM Tris, 0.1 M sodium chloride, 0.05% Tween20, pH = 7.2

## Preparation of recombinant glutaredoxin 1 and metallothionein 1X

The coding sequence of human GRX1 cDNA(NM_002064.3/ NP_002055.1, 1-106 residues) was cloned in the downstream of a His×6 tag coding sequence, between Xba-I and BamH-I sites of the pT7JLH plasmid[88]. Full size human MT1X cDNA (NM_005952.4/ NP_005943.2, 1-61 residues) was cloned into immediate downstream of T7 promoter, between Nde-I and BamH-I sites of the pT7JLH plasmid[88], to avoid addition of a His×6 tag coding sequence. Recombinant GRX1 was produced in *E. coli*, purified and carried out in vitro refolding for recombinant SENP1 with a His×6 tag.

MT1X was expressed in Rosetta strain of *E. coli* (Novagen) as above. The inclusion bodies of MT1X were washed twice with 2% Triton X-100, 0.1 M Tris, pH = 8, and then the inclusion bodies were dissolved in 6 M guanidine HCl, 10 mM DTT, 10 mM EDTA and 0.2 M Tris, pH = 8. The protein concentration was determined using BioRad Bradford protein assay (BioRad), and then adjusted to 1 mg/ml. For preparation of Zn²⁺-bound or Zn²⁺-free MT1X, two milliliter of the solution (2 mg MT1X) was used for each refolding with or without ZnCl₂.

To prepare Zn²⁺-bound MT1X, the MT1X solution was exchanged to 6 M guanidine HCl, 0.2 M arginine, 0.5 mM zinc chloride, 0.2 M Tris, pH = 7.2 using a centrifugal molecular cut off filter unit, Amicon Ultra-4 3 kDa MWCO (Millipore), and then the final volume was adjusted to 4 ml with the same solution. The MT1X solution was transferred into a Snakeskin 3.5 kDa MWCO Dialysis tubing (Thermo Fisher), and then the dialysis was carried out at 4 °C against three different solutions subsequently. The MT1X was dialyzed against 0.2 M arginine, 0.1 mM ZnCl₂, 0.2 M Tris, pH = 7.2 for 24 h, against 0.1 mM ZnCl₂, 0.1 M Tris, pH = 7.2 for 24 h, and then 50 mM Tris, pH = 7.2 for 48 h. The final dialysis buffer was changed to fresh buffer three times.

To prepare Zn²⁺-free MT1X, the volume of MT1X solution was adjusted to 4 mL with 6 M guanidine HCl, 10 mM DTT, 10 mM EDTA and 0.2 M Tris, pH = 8, and then was transferred into a Snakeskin 3.5 kDa MWCO Dialysis tubing. The refolding of Zn²⁺-free MT1X was carried out at 4 °C against four different solutions subsequently; against 0.2 M Arginine, 1 mM EDTA, 1 mM DTT, 0.2 M Tris, pH = 7.2 for 24 h, against 1 mM EDTA, 1 mM DTT, 0.1 M Tris, pH = 7.2 for 24 h,

against 0.1 mM DTT, 20 mM Tris, pH = 7.2 for 24 h, and against 20 mM Tris, pH = 7.2 for 24 h. The final dialysis buffer was changed to fresh buffer three times.

After refolding, MT1X was concentrated using Amicon Ultra-4 3 kDa MWCO to 200–300 μg/mL, and the filtrates of Amicon Ultra-4 were kept for using as negative buffer control for MT1X assays, particularly to confirm the removal of free zinc in MT1X solution.

### In vitro SENP1-protease assays

Two assays were developed to measure SENP1 activity. Recombinant His×6-SUMO1-mCherry was used as a substrate for both assays. The cDNA of full size human SUMO1, including the cleavage site, and mCherry were amplified with PCR using Phusion DNA polymerase and the PCR fragments were cloned into a pT7JLH plasmid at Xba-I site using Gibson assembly to make pT7JLH-SUMO1-mCherry plasmid. Transformed Rosetta strain of *E. coli* was grown in 50 ml LB broth containing 100 μg/ml Ampicillin at 37 °C with shaking until A600 reached between 0.6 and 0.8. His×6-SUMO1-mCherry protein was expressed by slow induction with 0.1 mM IPTG at 20 °C overnight and the *E. coli* pellet was harvested by centrifugation at 10,000 × g for 10 min. Pink color of *E. coli* pellet indicated the proper folding of His×6-SUMO1-mCherry. The pellet was incubated with 500 μg/ml lysozyme, 1 mM PMSF, 5 mM EDTA, 50 mM Tris, 0.2 M sodium chloride, pH = 8 for 3 h at 4 °C, and then sonicated five times in short burst of 30 s on ice. The cell lysate was centrifuged at 10,000 × g for 10 min at 4 °C, and the supernatant was dialyzed against 20 mM Tris, 0.3 M sodium chloride, pH = 8. The soluble His×6-SUMO1-mCherry was purified with a HisPur Nickel-NTA resin column (1 ×5 cm). The purified His×6-SUMO1-mCherry was dialyzed against 5 mM EDTA, pH = 8 for 6 h and then SENP1 assay buffer (20 mM Tris, 0.1 M sodium chloride, 0.05 % Tween20, pH = 7.2) for 24 h. The final dialysis buffer was changed to fresh buffer twice. The concentration of His×6-SUMO1-mCherry protein was estimated by the absorbance at 280 nm.

For the first assay, HisPur Nickel-NTA resin was used to capture and precipitate the undigested substrate, and then the mCherry signal in the supernatant containing the digested substrate was measured. 100 nM of recombinant SENP1 protein was mixed with 1 μM of His×6-SUMO1-mCherry, and incubated at 37 °C for 2 h to cleave mCherry from His×6 tag-SUMO1. The undigested substrate was captured and removed by the addition of 25 μl of HisPur Nickel-NTA resin suspension (1:1, v/v). After vortexing for 10 s, HisPur Nickel-NTA resin was precipitated by centrifugation at 10,000 × g for 30 s at 4 °C. Each supernatant was transferred into a 96 well clear bottom black plate and then fluorescent intensity, Ex544 nm/Em615 nm, was measured using Envision Multilabel plate reader (PerkinElmer) or Synergy HTX plate readers (Bio Tek Instruments).

The second assay was used to examine the effect of MT1X on SENP1 activity. Because MT1X interfered the binding of Hisx6-SUMO1-mCherry to HisPur Nickel-NTA resin, native-PAGE (non-denaturing Tris/Glycine gel electrophoresis) was used to separate the digested substrate from undigested substrate. Recombinant SENP1 protein without His×6 tag (200 nM) was mixed with 10 or 20 μg/mL (1.65 or 3.3 μM) MT1X, and 2.5 μM His×6-SUMO1-mCherry in a 20 μl reaction, incubated at room temperature for 45 min, and then separated by electrophoresis on non-denaturing Tris/Glycine gel. The mCherry signals of the undigested and digested substrate were detected using BioRad Gel Doc XR Imaging system (BioRad).

Because SENP1 activity changes during storage due to oxidation, the enzyme was fully activated prior to assays with 10 mM DTT at 4 °C overnight. Amicon Ultra-4 10 kDa MWCO was used to remove DTT and then replaced the buffer to SENP1 assay buffer, 20 mM Tris, 0.1 M sodium chloride, 0.05% Tween20, pH = 7.2. Final concentration of SENP1 and mutants were estimated by the absorbance at 280 nm. To preserve the activity of SENP1 25% glycerol (w/w) was added and stored under nitrogen gas at −20 °C.

### Analysis and statistics

Analysis was performed using FitMaster (HEKA Electronik) and GraphPad Prism (v10.1.1). Statistical outliers were identified and removed by an unbiased ROUT (robust regression followed by outlier identification) test. All data are shown as the mean ± SEM. Single-cell and single-islet data are shown as SuperPlots[89], with individual cells/islets in gray, and data averaged by animal as larger dark points. Statistical analysis was performed on data both with cells/islets as replicates, and on data averaged by animal/donor. Data were analyzed by the 2-tailed Student's *t* test (for two groups), or ANOVA and post-test as indicated in figure legends. A p-value less than 0.05 was considered significant. $Zn^{2+}$Bind (https://zincbind.net/) was used to predict the $Zn^{2+}$-binding site in SENP1 (2IYC)[90]. Free $Zn^{2+}$ for patch-clamp experiments was estimated with MaxChelator at https://somapp.ucdmc.ucdavis.edu/pharmacology/bers/maxchelator/.

### Reporting summary

Further information on research design is available in the Nature Portfolio Reporting Summary linked to this article.

### Data availability

RNA-seq data generated in this study have been deposited in the GEO database under accession code GSE249790. Other data generated in this study and used to generate figures are provided in the Source Data file. Source data are provided with this paper.

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

## Acknowledgements

The University of Alberta is situated on Treaty 6 territory, traditional lands of First Nations and Métis people. We thank the Human Organ Procurement and Exchange (HOPE) program and Trillium Gift of Life Network (TGLN) for their work in procuring human donor pancreas for research, and James Lyon (Alberta) for his efforts in human islet isolation. We especially thank the organ donors and their families for their kind gift in support of diabetes research. Work was funded by grants from the National Institutes of Health NIH R01 GM132610 (W.L.), Canadian Institutes of Health Research Foundation Grant 148451 (P.E.M.), and German Research Foundation TRR219 project M04, Project ID 322900939 (L.P.R.). H.L. was supported by a Sino-Canadian Studentship from Shantou University, A.R.P. holds the Canada Research Chair in Cell Therapies for Diabetes, and P.E.M. holds the Canada Research Chair in Islet Biology.

## Author contributions

Conceptualization: H.L., K.S., A.R.P., E.A., W.L., P.E.M. Methodology: H.L., K.S., L.N., L.P.R., A.R.P., E.A., W.L., P.E.M. Investigation: H.L., K.S., N.S., X.L., L.N., S.F., A.F.S., X.D., A.B., M.F., S.A., L.P.R. Visualization: H.L., P.E.M. Supervision: A.R.P., L.P.R., E.A., W.L., P.E.M. Writing—original draft: H.L., K.S., P.E.M. Writing—review & editing: H.L., K.S., N.S., X.L., L.N., S.F., A.F.S., X.D., A.B., M.F., S.A., A.R.P., L.P.R., E.A., W.L., P.E.M.

## Competing interests

The authors declare no competing interests.
