## [Peer Review File · Nature Communications]

β -cell responses to high fat feeding:
A role and mechanism for redox sensing by SENP1Editorial Note: Parts of this Peer Review File have been redacted as indicated to remove third-party material where no permission to publish could be obtained.

REVIEWER COMMENTS

Reviewer #1 (Remarks to the Author):

The manuscript by Lin, Suzuki and colleagues describes studies to investigate pancreatic beta cell functional responses to metabolic stress. Using a combination of in vitro studies of human cadaveric islets and mouse models they provide convincing evidence for key roles for SENP1 and a more reduced cytosolic redox state in the compensatory changes that result in enhanced GSIS in prediabetes.

This is a carefully performed study that uses state of the art methodology and derives conclusions that are well supported by the data. My only significant concern is a failure to adequately discuss possible gender specific differences in the mechanism they uncover. Thus there is a clear difference in the acute response of the male and female SENP1 KO animals to a high fat diet, but no data are provided as to whether human islets from overweight and/or diabetic subjects show a similar discordance, or if this is merely a species or strain dependent phenomenon. The gender compositions of the donors in each group for the experiments in Fig 1 are not provided making it impossible to determine how well they are matched for this critical variable. This should be discussed in greater detail and a summary table of the demographics of the human subjects in each group provided.

Reviewer #2 (Remarks to the Author):

The authors have investigated the functional adaptation that β -cells undergo upon short-term increase in demands for insulin. Specifically, they have characterized the role of redox signalling and SENP1 in the upregulation of β -cell function in mice fed a HFD for 48h. Loss of Senp1 activity (via inhibition or genetic ablation) results in loss of β -cell compensation to HFD.

While I think the study is highly interesting, there are a number of question marks that need

to be straightened out. I am most concerned about how some of the data is handled in the statistical analyses. have listed some comments to the authors below

1. Please provide a table summarizing the characteristics of the human donors
2. The statistical tests used are indicated in each figure legend, however, it is unclear to the reader which statistical tests have been used for which figure panel.
3. Figure 1B: it is unclear to me if the number of cells (63-236 and 21-36) is cells per donor or total number of cells analysed for non-diabetic donors and donors with T2D, respectively. And a "follow-up" question, is the statistical analysis performed on the donor data (i.e. 11-33 vs 4-5), or cells (i.e. 63-236 vs 21-36)?
4. Figure 1A and C: what is the "control" in the legend on the y-axis?
5. Figure 1B: please indicate in the results paragraph or figure legend that this is patch clamp experiments, otherwise the reader (at least I did until I came to the methods) will wonder if the cells are sorted and/how they were identified as β -cells.
6. If I understand correctly, figure 1A and 1B shows β -cell secretion at different glucose levels, one in the form of actual secretion (measurement of released insulin), and the other in the form of exocytosis (patch clamp measurements). However, I do not understand why you have significant differences between BMI <25 ND and BMI >25 ND at 10mM glucose in A but not in B (and vice versa at 1mM) but I am guessing this is due to the nature of the experiments. Regarding the significant difference at 1mM glucose in 1B you write "suggesting an increased insulin granule priming or amplifying effect even at low glucose". Would this not be detected also in the experiments in Fig 1A, and if there is priming/amplifying effects, would these not also affect secretion at high glucose? Please discuss/clarify.
7. Figure 1A-D: Please discuss potential reasons for why the data looks different in older humans
8. Figure 1H and supp figure 1C: the insulin content is expressed as a concentration (pg/ml), please convert this to an amount (pg/islet for example). Also, what are the dots representing? The number of dots does not seem to match the numbers in the respective legend
9. Figure S1D-E: you write 8+8 and 7+7 mice, but there are many more dots in the graphs. What are the dots representing?

10. Similar to comment 6, why do you not see significant differences in insulin secretion at low glucose (fig 1G) when you have increased exocytosis (fig 1I)? And vice versa at high glucose, why do you not see differences in exocytosis when you see differences in secretion?
11. Figure 2A: you write “n=216 genes, N=4, 3 mice”, does this mean that islets from 4 CD and 3 HFD mice were analysed? These are very small numbers and ideally should be increased.
12. There is no reference to the supplementary table in the paragraph on RNA seq, please add. And the table needs more information, at least mean expression and SD for both groups.
13. If I understand the methods section correctly you consider genes with $p < 0.1$ significant in the RNAseq data? And there is no correction for multiple testing. I feel this is not stringent enough, please provide a reasoning behind this or use stricter criteria. The p-value for your main candidate *Senp1* is an unimpressive 0.092. A larger number of animals in this analysis (see comment 12) will help.
14. Figure 2B: please provide the full list of significant pathways in a supp table, including information on which genes in these pathways that are differentially expressed.
15. Why did you use two different methods to measure oxygen consumption and why are they giving different results)? I.e., why may it be significant at 2.8mM glucose in B but not in A?
16. The oligomycin-sensitive respiration (i.e. the drop after addition of oligomycin) and maximum respiration looks quite different between the 2 groups. Can you do similar graphs as in 3B for those? Or specify in the text that there are not significant differences for these parameters? And irrespective of what you chose to do, please present the p-values for those comparisons in the figure or text.
17. Can you indicate the different respiratory parameters you measure in Figure 3A? For example, you write that respiration driving proton leak is higher in the 2-day HFD islets and those not familiar with mitochondrial respiration will probably believe that proton leak is simply the values that the data points indicate in the graph while in reality it's the difference between the values at the data points and the non-mitochondrial respiration.
18. Figure 3C: if I understand correctly, you have analysed these data as if you have $n=65+41$. This is not correct, you have $n=6+6$ (this is the same issue I tried to raise in

comment 2 above, if my point was unclear).

19. Figure 3D-E: Same as above; n=4+4 and 7+7, not 26-56 and 39-56

20. You write "(not shown)" for the upregulation of Senp1 in the RNAseq data, please refer to the supplementary table instead.

21. Figure 3G: you write N=2-9 mice, but there is no bar with less than 4 dots, and several bars have more than 9 dots. What do the dots represent?

22. Figure 3H: same as comment 14 and 15. This is true for several other figure panels both in the main file and in the supplementary. I will stop commenting on it, but in my view this way of analysing the data is not ok and needs to be changed throughout the manuscript.

23. Figure 4E-F: there is no indication of significance for CD WT mice at 2.8 mM compared to 10mM. Was it not significant?

24. For the GTTs you sometimes use 1g/kg and sometimes 0.5g/kg. Why?

25. Figure 5D: you write that infusion of WT SENP1 increased exocytosis but there is no indication of significance in the graph, please add (or are the 2 stars the significance of that comparison? If so, clarify).

26. In the discussion there are references to data not mentioned in the results section. Maybe Nat Com has other rules regarding this, but if you ask me all data mentioned in the discussion should be mentioned in the results section.

27. In the methods you mention ITTs, but there is no ITT data in the manuscript

28. In the methods section on in vitro Senp1 assays you refer to Fig. S2A which is an IPGTT figure.

Reviewer #3 (Remarks to the Author):

The authors have explored molecular mechanisms affecting pancreatic beta-cell insulin secretion in response to metabolic stress. Using in vitro cell based assays, they demonstrate that insulin secretion and exocytosis is enhanced in beta-cells from overweight humans and mice fed a 2-day high-fat diet without increased calcium influx. RNA-Seq analysis of mouse beta-cells isolated following a 2-day high fat diet revealed early changes in metabolic pathways and increases expression of the SENP1 SUMO protease. Enhanced insulin secretion and exocytosis following a 2-day high-fat diet was found to be dependent reduced intracellular redox. In addition, the SENP1 SUMO protease was found to be required using

SENP1 pancreas and beta-cell knockout mice. In vitro assays using recombinant SENP1 catalytic domain revealed that redox sensitivity was dependent on C535 and modulated by Zn⁺. Cell-based assays supported roles for C535 and Zn⁺ in affecting SENP1 activity and its role in affecting beta-cell exocytosis and insulin secretion.

Previous studies by this group and others have supported roles for the SENP1 SUMO protease in regulating insulin-secretion by beta-cells. Previous studies have also indicated that SENP1 activity is subject to redox regulation. The specific molecular mechanisms by which SENP1 controls beta-cell function and how this may be impacted by changes in redox potential of the cells, however, has remained uncertain. The studies reported in this manuscript address these important questions and provide new and potentially valuable insights.

Overall, the manuscript was a challenge to read and understand. This could be addressed by more clearly explaining specific questions underlying individual experiments and putting the results in the context of these questions. Moreover, more background information and context is needed to appreciate the significance of experiments looking at exocytosis and insulin secretion. Are the experiments particularly novel and significant because of the short, 2-day, high-fat diet? Or because the experiments are looking specifically at exocytosis? For someone not in the diabetes field, it is unclear how novel these experiments are. Have these specific questions not already been addressed in other studies? What makes the specifics of the RNA-Seq experiment novel?

A second major concern is the complete absence of any discussion of sumoylation. It is unclear whether the authors think that SENP1 is functioning as a SUMO protease to affect beta-cell function, or in some other capacity. Some discussion is needed. Experiments evaluating effects of SENP1 on overall sumoylation levels in experiments involving introduction of recombinant proteins would also be very informative. Do effects or lack of effects of SENP1 on exocytic capacity and insulin secretion correlate with effects on sumoylation?

More specific comments are as follows:

(1) In Figure 1A, insulin release is similar in all three groups tested at 1mm glucose. In Figure 1B, exocytic capacity is shown to be higher in the BMI > 25 ND group at 1 mm glucose. Why do exocytic capacity and glucose release not correlated at 1 mm glucose?

(2) For many experiments (1A-D, 4E and F...), there is no data collected at 0 glucose. In 4E and F this is a concern because it is concluded that loss of SENP1 prevents up-regulation of exocytosis. Without a baseline of exocytosis in the absence of glucose, it is unclear if there is no up-regulation, or if the level of upregulation is decreased.

(3) Intracellular dialysis or infusion is used to introduce recombinant SENP1 proteins into cells in multiple experiments. The authors should provide an explanation of this technique and details of SENP1 expression. What is the efficiency of the technique in terms of amount of protein delivered, and fraction of cells taking up the protein? In addition, it would be helpful to have information of the intracellular localization of the SENP1 protein. The N-terminus of SENP1 is known to contain determinants of nuclear localization. Where does the catalytic domain concentrate in the beta-cells? Also, as discussed above, possible effects on introducing recombinant proteins into cells on sumoylation levels would be informative.

(4) What is GRX1 and why does it influence the effect of GSH on SENP1 activity (Figure 5E)?

(5) In Figure 5D, the authors switch to human beta-cells. It would be more informative to see this experiment done in beta-cells isolated from WT and betaSENP1 KO cells.

(6) The results of Figure 5B and 5C are confusing. First, the authors show that the 535S mutant is 2-fold more active and then they show that it has the same activity. There is some mention that "some recombinant proteins were likely partially inactivated by oxidation". This section needs to be cleaned using proteins of similar quality.

(7) The proton transfer pathway (Figure 6SB) to explain SENP1 redox-sensing is intriguing, but needs additional support. In particular, do mutations of H533 or D550 support this model?

RESPONSE TO REVIEWER COMMENTS

Reviewer #1 (Remarks to the Author):

The manuscript by Lin, Suzuki and colleagues describes studies to investigate pancreatic beta cell functional responses to metabolic stress. Using a combination of in vitro studies of human cadaveric islets and mouse models they provide convincing evidence for key roles for SENP1 and a more reduced cytosolic redox state in the compensatory changes that result in enhanced GSIS in prediabetes.

This is a carefully performed study that uses state of the art methodology and derives conclusions that are well supported by the data. My only significant concern is a failure to adequately discuss possible gender specific differences in the mechanism they uncover. Thus there is a clear difference in the acute response of the male and female SENP1 KO animals to a high fat diet, but no data are provided as to whether human islets from overweight and/or diabetic subjects show a similar discordance, or if this is merely a species or strain dependent phenomenon. The gender compositions of the donors in each group for the experiments in Fig 1 are not provided making it impossible to determine how well they are matched for this critical variable. This should be discussed in greater detail and a summary table of the demographics of the human subjects in each group provided.

Response: We thank the reviewer for this comment and we have now performed additional analyses of human islet insulin secretion and beta-cell exocytosis data by sex (shown in **Suppl Figure 1A-H**). We do indeed see a similar sex-dependent difference in the human data, as reported in the mice, and now discuss this in the manuscript text (page 4, page 10). We note however, although these trends seem clear, that the separation of human data by sex reduces the power for analysis somewhat. Details on donor characteristics, including sex, are now provided in a revised **Supplementary Table 1**.

Reviewer #2 (Remarks to the Author):

The authors have investigated the functional adaptation that β -cells undergo upon short-term increase in demands for insulin. Specifically, they have characterized the role of redox signalling and SENP1 in the upregulation of β -cell function in mice fed a HFD for 48h. Loss of Senp1 activity (via inhibition or genetic ablation) results in loss of β -cell compensation to HFD.

While I think the study is highly interesting, there are a number of question marks that need to be straightened out. I am most concerned about how some of the data is handled in the statistical analyses. have listed some comments to the authors below

1. Please provide a table summarizing the characteristics of the human donors

Response: We now provide this information in a revised **Supplementary Table 1**.

2. The statistical tests used are indicated in each figure legend, however, it is unclear to the reader which statistical tests have been used for which figure panel.

Response: We have now indicated which statistical tests are used for which panel in all figure legends.

3. Figure 1B: it is unclear to me if the number of cells (63-236 and 21-36) is cells per donor or total number of cells analysed for non-diabetic donors and donors with T2D, respectively. And a "follow-up" question, is the statistical analysis performed on the donor data (i.e. 11-33 vs 4-5), or cells (i.e. 63-236 vs 21-36)?

Response: The number of cells for the exocytosis experiments indicated the **total** number of cells studied, and statistics were originally performed on cells (not donors) for these single-cell measurements (as typical in the field, but **see also response to comment 22, below, which addresses this in detail**). We have now: **1.** Clarified the range of cells and total donors per condition in the figure legends; and **2.** Provide a statistical analysis both 'by cells' and 'averaged per donor', and present the latter in **Suppl Fig 1I-N**.

4. Figure 1A and C: what is the “control” in the legend on the y-axis?

Response: We thank the reviewer for pointing out this typo. It should read “% of Content” (not ‘control’). This has now been corrected.

5. Figure 1B: please indicate in the results paragraph or figure legend that this is patch clamp experiments, otherwise the reader (at least I did until I came to the methods) will wonder if the cells are sorted and/how they were identified as β -cells.

Response: This is now indicated in both the results (page 4) and figure legend for Figure 1.

6. If I understand correctly, figure 1A and 1B shows β -cell secretion at different glucose levels, one in the form of actual secretion (measurement of released insulin), and the other in the form of exocytosis (patch clamp measurements). However, I do not understand why you have significant differences between BMI <25 ND and BMI >25 ND at 10mM glucose in A but not in B (and vice versa at 1mM) but I am guessing this is due to the nature of the experiments. Regarding the significant difference at 1mM glucose in 1B you write “suggesting an increased insulin granule priming or amplifying effect even at low glucose”. Would this not be detected also in the experiments in Fig 1A, and if there is priming/amplifying effects, would these not also affect secretion at high glucose? Please discuss/clarify.

Response: The reviewer is correct that this difference is due to the nature of the experiment. In ‘A’ we are measuring ‘insulin secretion stimulated by increasing concentrations of glucose’ while in ‘B’ we are measuring ‘beta-cell exocytosis stimulated by direct membrane depolarization, performed at the ambient glucose concentration indicated’. The latter single-cell studies are somewhat analogous to measuring insulin secretion in response to depolarizing KCl at low glucose (see the studies of JC Henquin in the 1990’s – e.g. PMID: 8383702). Thus, the increase in ‘depolarization stimulated exocytosis’ at low glucose in the overweight humans or HFD mice is indicative of the ‘size of the pool of granules that can respond to a depolarizing stimulus’ (in the insulin secretion measurements at low glucose in Fig 1A, although there is a larger pool of insulin granules to release in the overweight humans....there is still no action potential firing at low glucose...so the increased secretion in comparison to lean individuals only manifests once glucose is raised). I agree that we sometimes have difficulty in clearly conveying this overall concept and have attempted to clarify in text throughout (see page 4 and page 9). See also response to comment 10, below, where I’ve tried for a clearer explanation.

7. Figure 1A-D: Please discuss potential reasons for why the data looks different in older humans

Response: We have now added a very brief comment on this (page 4-4), and refer the readers to our recently published work on insulin secretion and beta-cell phenotypes during ageing (PMID: 36197983).

8. Figure 1H and supp figure 1C: the insulin content is expressed as a concentration (pg/ml), please convert this to an amount (pg/islet for example). Also, what are the dots representing? The number of dots does not seem to match the numbers in the respective legend

Response: We have now expressed Fig 1H and Suppl Fig 1C (now Suppl Fig 2C) as pg/islet. The points represent measurements from individual animals and we have now corrected the figure legend (and checked all others throughout, correcting where necessary).

9. Figure S1D-E: you write 8+8 and 7+7 mice, but there are many more dots in the graphs. What are the dots representing?

Response: As above, dots represent measurements from individual mice. As above, we have checked and corrected throughout. In this particular instance experiments in Fig S1D, E (now Fig S2D, E) are from 8 Chow fed and 8 HFD (corrected to 7 CD and 8 HFD; now Fig S2D) or 7 Chow fed and 7 HFD (S2E) mice. We have clarified numbers (of cells, islets, and animals as appropriate) throughout the figure legends.

10. Similar to comment 6, why do you not see significant differences in insulin secretion at low glucose (fig 1G) when you have increased exocytosis (fig 1I)? And vice versa at high glucose, why do you not see differences in exocytosis when you see differences in secretion?

Response: *The response here is the same as to Comment 6. In brief, the insulin secretion measurements are reporting 'glucose dependent insulin secretion' while the single-cell exocytosis measurements are reporting the 'exocytotic capacity' at a given glucose concentration (that 'capacity' is simply not released at low glucose in the insulin secretion measurements, without some other stimulus like KCl or high-glucose). I find this to be a very interesting concept, since it tells us that 'something' (we think redox control via SENP1, as in the rest of the paper) is acting to increase the 'releasable' pool of insulin granules, even at low glucose, when animals are given a short term HFD, and this occurs BEFORE glucose increases (at which time the 'increased capacity' is tapped into to allow increased insulin secretion). Again, I have tried to clarify this in the revised text (page 4).*

As for the second part of this question, this also comes down to how the experiments were performed. In the exocytosis measurements, the cells were in a bath solution with the noted glucose concentration (i.e. bathed in a given glucose before and during the patch-clamp). The response here indicates the 'exocytotic capacity' at a given prevailing glucose concentration (and stimulation was by strong patch-clamp depolarization). Conversely, the insulin secretion measurements show the secretory response to stimulation with a given glucose concentration. Thus, the exocytosis measurements at 10 mM glucose were likely not different because that 10 mM glucose was 'ramping up' exocytotic capacity (and filling the granule pool, perhaps through similar mechanisms as with the HFD at low glucose) before stimulation. While the glucose-stimulated secretion was 'acting on' the granule pool that was set prior to the glucose-stimulation. I hope that our attempts to clarify in text have made this a bit clearer.

11. Figure 2A: you write "n=216 genes, N=4, 3 mice", does this mean that islets from 4 CD and 3 HFD mice were analysed? These are very small numbers and ideally should be increased.

Response: *The reviewer is correct that these were from islet preparations (followed by sorting of beta-cells) from 4 and 3 mice. We were unfortunately unable to repeat these experiments, as the personnel responsible for these experiments have moved on. I will point out that RNAseq of cells from 3-4 mice are not uncommon (e.g. PMID: 28380380). We have however understand the reviewers point, particularly with respect to genes with nominal significance, and we have therefore de-emphasized discussion of individual genes from this work and removed reference to Senp1 and metallothionein based on these data - see below).*

12. There is no reference to the supplementary table in the paragraph on RNA seq, please add. And the table needs more information, at least mean expression and SD for both groups.

Response: *I believe that the original reference was to 'Data S1', but this original data table has now been expanded and re-labeled as **Supplementary Table 2**, and referred to on pages 5 and 14.*

13. If I understand the methods section correctly you consider genes with $p < 0.1$ significant in the RNAseq data? And there is no correction for multiple testing. I feel this is not stringent enough, please provide a reasoning behind this or use stricter criteria. The p-value for your main candidate Senp1 is an unimpressive 0.092. A larger number of animals in this analysis (see comment 12) will help.

Response: *The reviewer is correct and, given that we have not increased our numbers, we have removed the discussion of Senp1 in this context. We have however retained the pathways analyses in Figure 2 since these highlighted pathways with a level of significance that is much more convincing (e.g. $P < 10^{-4}$ - 10^{-6}). We do maintain however, the Senp1 expression measured by qPCR (Fig 3G).*

14. Figure 2B: please provide the full list of significant pathways in a supp table, including information on which genes in these pathways that are differentially expressed.

Response: *We have now included this as **Supplementary Table 3** (now referenced on pages 5 and 15).*

15. Why did you use two different methods to measure oxygen consumption and why are they giving different results? I.e., why may it be significant at 2.8mM glucose in B but not in A?

Response: In Figure 3A we measure oxygen consumption using the Seahorse assay following 2-day HFD. Although this suggested an increased basal (2.8 mM glucose) oxygen consumption, the results did not reach statistical significance (we now show the calculated basal OCR and P-value ($p=0.06$) in Revised Fig 3A). To look at basal oxygen consumption in isolation, and since that basal condition was directly relevant to the increased exocytosis we see at 2.8 mM glucose, and tested basal OCR alone using another method (fluorescence lifetime imaging) and find a significant increase in basal OCR with this after 2-day HFD (Fig 3B). We have now elaborated on this in the manuscript text (page 6).

16. The oligomycin-sensitive respiration (i.e. the drop after addition of oligomycin) and maximum respiration looks quite different between the 2 groups. Can you do similar graphs as in 3B for those? Or specify in the text that there are not significant differences for these parameters? And irrespective of what you chose to do, please present the p-values for those comparisons in the figure or text.

Response: We have now provided direct comparison of the different conditions from the Seahorse OCR data (revised Fig 3A). In this experiment, basal OCR does not quite reach statistical significance ($P=0.06$), while both proton leak and non-mitochondrial OCR are significantly increased in the 2-day HFD ($P<0.01$), although the latter is a small but significant difference. The maximum respiration also trends towards statistical significance ($P=0.09$) and we now show that P-value suggested by the reviewer.

17. Can you indicate the different respiratory parameters you measure in Figure 3A? For example, you write that respiration driving proton leak is higher in the 2-day HFD islets and those not familiar with mitochondrial respiration will probably believe that proton leak is simply the values that the data points indicate in the graph while in reality it's the difference between the values at the data points and the non-mitochondrial respiration.

Response: The reviewer is correct, and we have now added an indication of these parameters (coloured boxes, and italicised labels), along with a comparison of individual parameters (noted by 'i-vi').

18. Figure 3C: if I understand correctly, you have analysed these data as if you have $n=65+41$. This is not correct, you have $n=6+6$ (this is the same issue I tried to raise in comment 2 above, if my point was unclear).

Response: The reviewer is correct. And I agree that this data should be analyzed by animal (not by islet). To me, this is indeed clear-cut (as opposed to the single-cell electrophysiology...see response to comment 22, below). This has now been corrected.

19. Figure 3D-E: Same as above; $n=4+4$ and $7+7$, not 26-56 and 39-56

Response: Please see detailed response to comment 22, below.

20. You write "(not shown)" for the upregulation of Senp1 in the RNAseq data, please refer to the supplementary table instead.

Response: As per our response to Comment 13, this has now been removed (and we instead refer to the qPCR data).

21. Figure 3G: you write $N=2-9$ mice, but there is no bar with less than 4 dots, and several bars have more than 9 dots. What do the dots represent?

Response: This has now been corrected (and other figures/legends have been checked throughout). Dots now refer to individual mice. Thank you for pointing out this error.

22. Figure 3H: same as comment 14 and 15. This is true for several other figure panels both in the main file and in the supplementary. I will stop commenting on it, but in my view this way of analysing the data is not ok and needs to be changed throughout the manuscript.

Response: This is a very interesting point, and my response here can be boiled down to: ‘this is how single-cell electrophysiology has always been done’...but ‘the reviewer has a point, and we should try to do better’ with revision and edits done accordingly.

Firstly, the reviewer is correct that all single-cell electrophysiology (i.e. exocytosis) has been analyzed ‘by cells’ (not by averaging cells for each animal). Largely, this is how the field has always done it – and to be honest it is hard, if not impossible, to find an example otherwise. Most (perhaps all) single-cell electrophysiology is analyzed in this way. Some recent examples can be seen in the islet field (e.g. PMID: 35997256; PMID: 37494670; PMID: 30187652; PMID: 31358556; PMID: 34345920; PMID: 37058408), or in neuroscience (e.g. PMID: 37566700; PMID: 37548854; PMID: 37536499; PMID: 37158590) -- including islet cell work published in this journal (PMID: 35869052; PMID: 36309517).

That being said, the reviewer’s point is well taken. And history is not a good reason to maintain our previous practice. Thus, we now present all single-cell electrophysiology in the paper as SuperPlots – as described here (PMID: 32346721). In this way, we show all cells (background grey) but focus on data averaged by animal (larger, dark, dots). Similarly, analyses are now presented both for ‘analysis by cells’ and ‘analysis by animal’. The result is shown below (Using Fig 3E as an example, where significance is indicated on the cell-level by ‘@’, and on the animal level by ‘*’):

This change has been made throughout the paper when reporting single-cell function data. I hope the Reviewer can appreciate that we are trying to strike a balance between being responsive to this valid concern, while maintaining some link to the norms of how this data is typically presented. Personally, I think it is an improvement that shows animal-level data while maintaining information on the cell-variability and I wish to thank the Reviewer for raising the point.

23. Figure 4E-F: there is no indication of significance for CD WT mice at 2.8 mM compared to 10mM. Was it not significant?

Response: As mentioned in response to the above point (22), we have now re-done all exocytosis figures and analysis (we initially didn’t compare the 2.8 vs 10 mM glucose). As shown in revised Figs 4E, the 2.8->10mM glucose response is significant at the ‘analysis by cells’ and ‘analysis by animals’ levels. At the 4 week time point this doesn’t quite reach statistical significance.

24. For the GTTs you sometimes use 1g/kg and sometimes 0.5g/kg. Why?

Response: This is an important question, and we regret that we did not explain it sufficiently in the manuscript. Experiments were performed initially, in the absence of any HFD, with 1g/kg. After HFD we found that in some animals/conditions (particularly 4 week HFD) 1g/kg was too high. Because of the development of glucose intolerance, the glucose excursions were routinely ‘maxing-out’ the glucometer measurements. We therefore reduced to 0.5g/kg under those conditions. This has now been clarified in the manuscript text (page 14).

25. Figure 5D: you write that infusion of WT SENP1 increased exocytosis but there is no indication of significance in the graph, please add (or are the 2 stars the significance of that comparison? If so, clarify).

Response: As mentioned in response to the above point (22), we have now re-done all exocytosis figures and analysis. With respect to the original Fig 5D (which is now Fig S7D) we now indicate significance following analysis with cells as replicates ($P < 0.01$) or with cells averaged by human donor ($P < 0.05$). Similarly, significance with SENP1 infusion in the replacement figure (in SENP1-KO mice, new Fig. 5F) is also now indicated.

Please note, in some experiments in Figure 5, we present statistical comparisons 'within groups' (e.g. new Fig. 5D) reflecting the comparisons of interest. This has been clearly indicated in the figure legend.

26. In the discussion there are references to data not mentioned in the results section. Maybe Nat Com has other rules regarding this, but if you ask me all data mentioned in the discussion should be mentioned in the results section.

Response: We have removed all reference to 'data not shown', which upon our review in response to this comment were not really essential and distracted from the main message.

27. In the methods you mention ITTs, but there is no ITT data in the manuscript

Response: The reviewer is correct that the paper has no ITT data. Inclusion of this resulted from the use of our general methods template for our in vivo work in generating this section for the paper, and reference to ITT has now been removed from the Methods section.

28. In the methods section on in vitro Senp1 assays you refer to Fig. S2A which is an IPGTT figure.

Response: Thank you for pointing this out. This must have been left from an earlier version of the paper and has now been removed.

Reviewer #3 (Remarks to the Author):

The authors have explored molecular mechanisms affecting pancreatic beta-cell insulin secretion in response to metabolic stress. Using in vitro cell based assays, they demonstrate that insulin secretion and exocytosis is enhanced in beta-cells from overweight humans and mice fed a 2-day high-fat diet without increased calcium influx. RNA-Seq analysis of mouse beta-cells isolated following a 2-day high fat diet revealed early changes in metabolic pathways and increases expression of the SENP1 SUMO protease. Enhanced insulin secretion and exocytosis following a 2-day high-fat diet was found to be dependent reduced intracellular redox. In addition, the SENP1 SUMO protease was found to be required using SENP1 pancreas and beta-cell knockout mice. In vitro assays using recombinant SENP1 catalytic domain revealed that redox sensitivity was dependent on C535 and modulated by Zn⁺. Cell-based assays supported roles for C535 and Zn⁺ in affecting SENP1 activity and its role in affecting beta-cell exocytosis and insulin secretion.

Previous studies by this group and others have supported roles for the SENP1 SUMO protease in regulating insulin-secretion by beta-cells. Previous studies have also indicated that SENP1 activity is subject to redox regulation. The specific molecular mechanisms by which SENP1 controls beta-cell function and how this may be impacted by changes in redox potential of the cells, however, has remained uncertain. The studies reported in this manuscript address these important questions and provide new and potentially valuable insights.

Overall, the manuscript was a challenge to read and understand. This could be addressed by more clearly explaining specific questions underlying individual experiments and putting the results in the context of these questions. Moreover, more background information and context is needed to appreciate the significance of experiments looking at exocytosis and insulin secretion. Are the experiments particularly novel and significant because of the short, 2-day, high-fat diet? Or because the experiments are looking specifically at exocytosis? For someone not in the diabetes field, it is unclear how novel these experiments are. Have these specific questions not already been addressed in other studies? What makes the specifics of the RNA-Seq experiment novel?

Response: We thank the reviewer for their time and effort in assessing this work and providing constructive suggestions. We have edited the text throughout in an effort to increase readability, to clarify the questions

being addressed, and to better explain significance (see text in RED throughout the revised version). Briefly, metabolic demand changes depending on many factors...fasting, overnutrition, exercise...and the pancreas must adapt its insulin output accordingly. While much is known about 'triggers' for insulin release (i.e. a rise in blood glucose stimulating Ca²⁺ responses in beta-cells), very little is known about how beta-cells adapt the 'magnitude' of such responses to match peripheral demand (in rodents, this occurs over weeks by increasing islet size...but this is unlikely to be the case in adult humans where islet growth is limited). This is important, because a failure to adapt appropriately to increased metabolic demand (i.e. by increasing insulin output with overnutrition or obesity) is a key factor in the development of diabetes. We show that: **1.** An increased 'capacity' to secrete insulin (here defined as 'exocytotic capacity') occurs very quickly after high fat feeding in mice (and we present evidence that the same happens in 'young-ish' humans); **2.** this requires redox signalling via SENP1 to augment 'exocytotic capacity', even before a glucose stimulus; and **3.** this is important for maintaining normal glucose homeostasis early in response to high fat feeding. I hope that the edits to the paper have made these points clearer.

A second major concern is the complete absence of any discussion of sumoylation. It is unclear whether the authors think that SENP1 is functioning as a SUMO protease to affect beta-cell function, or in some other capacity. Some discussion is needed. Experiments evaluating effects of SENP1 on overall sumoylation levels in experiments involving introduction of recombinant proteins would also be very informative. Do effects or lack of effects of SENP1 on exocytic capacity and insulin secretion correlate with effects on sumoylation?

Response: We thank the reviewer for pointing out that this was unclear. While trying not to take too much space, we have expanded upon a previous statement in the introduction (red text, page 3) to indicate that several targets have been shown to regulate insulin exocytosis based on their SUMOylation status. At least two of these (synaptotagmin VII and tomosyn 1) are deSUMOylated upon glucose-stimulation of islets or beta-cells.

The question of course is whether "introduction of recombinant proteins" results in altered SUMOylation status of relevant targets. As mentioned, however, in response to a question 3 below we are directly infusing these recombinant SENP1 proteins into single-cells...as such we are unable to assess the SUMOylation status of specific targets in these particular experiments. We now, however, provide some additional discussion (page 10) to highlight several things:

1. the SUMO protease activity of SENP1 is required for its effect on exocytosis (shown in what is now Supplementary Figure 7A)
2. the inhibitory effects of intracellular dialysis of either SUMO1 or the SUMO-ligase (Ubc9) on exocytosis can be reversed by SENP1 (Dai et al., Diabetes, 2011). We've also shown that a conjugation deficient SUMO1 (Δ GG) has no effect on exocytosis.
3. SENP1 deSUMOylates the exocytotic target synaptotagmin VII (Dai et al., Diabetes, 2011)

More specific comments are as follows:

(1) In Figure 1A, insulin release is similar in all three groups tested at 1mm glucose. In Figure 1B, exocytic capacity is shown to be higher in the BMI > 25 ND group at 1 mm glucose. Why do exocytic capacity and glucose release not correlate at 1 mm glucose?

Response: This important question was also raised by Reviewer 2 (comments 6 and 10, above). In brief, the insulin secretion measurements are reporting 'glucose dependent insulin secretion' while the single-cell exocytosis measurements are reporting the 'exocytotic capacity' at a given glucose concentration (that 'capacity' is simply not released at low glucose in the insulin secretion measurements, without some other stimulus like KCl). What this means is that, in the young higher BMI donors (and the mice fed HFD), the 'capacity' of a beta-cell to mount a secretory response is increasing under low glucose conditions, prior to stimulation with increased glucose. This 'early preparation' likely contributes to the actual increase in glucose dependent insulin secretion once stimulated with high glucose. I hope that revisions to the manuscript text (pages 4 and 9) have made this clearer.

(2) For many experiments (1A-D, 4E and F...), there is no data collected at 0 glucose. In 4E and F this is a concern because it is concluded that loss of SENP1 prevents up-regulation of exocytosis. Without a baseline of exocytosis in the absence of glucose, it is unclear if there is no up-regulation, or if the level of upregulation is decreased.

Response: Experiments at 0 glucose are typically considered to be a bad idea in the islet biology field given the stress it imposes (see for example PMID: 14651961; PMID: 14651961). As for 'baseline', the threshold for glucose-dependent insulin secretion is generally between 3-5 mM glucose (in humans), and a bit above 5 mM glucose (in mice). This is why we use slightly different 'baseline' glucose concentrations in human as

Editorial note:
figure redacted

opposed to mouse studies. As shown at the left (glucose concentration-response from PMID: 26389676) both glucose concentrations should be sufficient for 'baseline' responses (at 1 or 2.8 glucose for example, we are largely unable to suppress exocytosis responses any further, including by infusion of SUMO1/Ubc9). In these figures (revised Fig 4E, F) I think that the 'upregulation of exocytosis at the baseline condition in the WT animals is seen (red arrow, in panel at right) while it is lost in the KO (blue arrow, panel at right) and from our experience the baseline will not get much lower.

(3) Intracellular dialysis or infusion is used to introduce recombinant SENP1 proteins into cells in multiple experiments. The authors should provide an explanation of this technique and details of SENP1 expression. What is the efficiency of the technique in terms of amount of protein delivered, and fraction of cells taking up the protein? In addition, it would be helpful to have information of the intracellular localization of the SENP1 protein. The N-terminus of SENP1 is known to contain determinants of nuclear localization. Where does the catalytic domain concentrate in the beta-cells? Also, as discussed above, possible effects on introducing recombinant proteins into cells on sumoylation levels would be informative.

Response: For this we include recombinant protein within the patch-clamp pipette, and during whole-cell electrophysiology the protein is dialyzed directly into the cell. This is common approach used with small molecules, proteins, and antibodies (a nice, older, example using antibodies is here: PMID: 9478993). In beta-cells which are quite small and uniformly spherical (in culture) compared with neurons, this dialysis occurs very quickly. For example, intracellular dialysis of purified SUMO1 peptide and/or Ubc9 completely suppresses exocytosis in beta-cells within a minute or two (...the fastest time point we measured; PMID: 21266332). One assumption of is that the concentration equilibrates rapidly as well, although we now point out this assumption as a caveat of these experiments (page 13). Details on the preparation of recombinant proteins is presented starting on page 16 ("Preparation of recombinant wild-type and mutant SENP1").

Given that the pipette solution volume is much (MUCH) larger than the cell volume, it is also assumed that the distribution of the protein will be relatively uniform throughout the cytosol (page 13). However, given the way that these experiments are performed (intracellular dialysis of the cell contents; and the rapidness with which effects are measured), the effects on exocytosis must be occurring through direct effects at or near the plasma membrane (the timing, and 'wash out' of cytosol, makes effects via mitochondria or gene transcription unlikely).

Because we are patch-clamping individual cells and directly dialyzing peptides into them, the 'efficiency of delivery' is 100%. And although we assume a uniform distribution of the catalytic domain peptides within the time frame studied, I suppose over time they may accumulate in distinct subcellular compartments. Again however, the experiments are performed over a very short timeframe (minutes). Of potential interest to the reviewer, we previously expressed fluorescently tagged full-length SENP1 within an insulin-secreting cell line (INS1 cells) and found that this targets to plasma membrane granules in these cells, and much less to the nucleus than expected (see expression at the plasma membrane in the TIRF images in Fig 4 of PMID: 26389676, while expression of SENP1-GFP in HEK293 cells was as expected - enriched in the nucleus). This has now been noted in the introduction (page 3).

In these single cell experiments (where we 'inject' or 'dialyze' the SENP1 catalytic peptides directly into individual cells) it will not be possible for us to measure SUMOylation (perhaps some future imaging-based assays would be useful here, but that is not feasible for us in a reasonable timeframe). Note however, that

we have previously performed extensive analysis with SUMO1 mutants (e.g. SUMOdeltaGG), ligases (e.g. Ubc9), and other mutants that have established a role for SUMOylation in downstream processes controlling insulin secretion (i.e. exocytosis: PMID: 21266332; PMID: 24970137; PMID: 26389676; PMID: 28325894) and others have published data confirming this (PMID: 30814610; PMID: 29299635).

Finally, as noted above, we and others have measured SUMOylation of several exocytotic proteins (synaptotagmin VII, syntaxin 1A, syntaxin 1A, RIM1a, tomosyn 1...) and at least one of these has been shown to be deSUMOylated by SENP1 (synaptotagmin VII). Now discussed in the introduction (page 3).

We hope that revision to the manuscript text have made some of these issues clearer.

(4) What is GRX1 and why does it influence the effect of GSH on SENP1 activity (Figure 5E)?

Response: Thank you for this question. Glutaredoxin 1 (GRX1) is a redox enzyme that uses reduced glutathione (GSH) as a co-factor for the regulation of several cellular processes – most notably in antioxidant defense (PMID: 25749165). It can also play an important role in the redox-dependent regulation of cell signalling (by breaking disulfide bonds, reducing thiol groups on target proteins: PMID: 30639960). With respect to insulin secretion, GRX1 has been shown to mediate the effects of reducing equivalents to augment beta-cell exocytosis (PMID: 15983215; PMID: 19299446) and we further demonstrated that GRX1 increases the efficiency of SENP1 activation by GSH (see Fig 5A in PMID: 26389676). We regret that important background was omitted and have now elaborated on this in the manuscript text (page 6, and bottom of page 7).

(5) In Figure 5D, the authors switch to human beta-cells. It would be more informative to see this experiment done in beta-cells isolated from WT and betaSENP1 KO cells.

Response: We have now performed the suggested experiments in SENP1 KO beta cells (with WT beta-cells as a control) (**new Fig 5F**). Because of the number of groups included in this experiment (the original in human experiment was a large undertaking) we have focused on the KO beta cells (with WT used as a control). We show that both SENP1 WT and SENP1 C535S both rescue the exocytosis defect (at 5 mM glucose) in the SENP1 KO beta-cells; and furthermore that H₂O₂ reverses that rescue only in the cells dialyzed with SENP1 WT, and not SENP1 C535S – suggesting again that the C535S is resistant to redox-dependent inhibition. This essentially replicates what we saw in the human cells (but without endogenous SENP1 present of course), and that data is now moved to **Supplementary Figure 7A**.

(6) The results of Figure 5B and 5C are confusing. First, the authors show that the 535S mutant is 2-fold more active and then they show that it has the same activity. There is some mention that “some recombinant proteins were likely partially inactivated by oxidation”. This section needs to be cleaned using proteins of similar quality.

Response: This is a good question and reflects our ‘learning curve’ in figuring out what was going on in these experiments. The purified recombinant enzymes become oxidized over time by exposure to air and show reduced activity...except for the C535S mutants. In Fig 5B, it is not so much that the C535S activity has ‘increased’ 2x, but rather that others have been gradually inactivated by oxidation while the C535S is resistant (to some extent this is also seen in the original Fig 5E - now Fig 5G - where the WT control is lower than C535S). Subsequently, experiments were performed with ‘reactivation’ by brief reduction with DTT (if we do that in Fig 5B for example, everything – except C603S – will have high activity). In short, I don’t think this is a ‘protein quality issue’ but rather it is indicative of the underlying mechanism we are studying. We hope to have clarified this (page 7) and, together with the new data provided (see point 7 below), that this is now presented more comprehensively.

(7) The proton transfer pathway (Figure 6SB) to explain SENP1 redox-sensing is intriguing, but needs additional support. In particular, do mutations of H533 or D550 support this model?

Response: We have now performed two sets of additional experiments with new SENP1 mutants. In **new Fig 5C**, we show that mutations of either H533 or D550 suppress SENP1 activity, but the suppressive effect of the latter (D550S) is partially rescued in the C535S mutant. This would be consistent with a suppressive effect of C535 via H533 (and interaction with D550). Furthermore, in **new Fig 5D**, we show that excess histidine (to compete with H535) also prevents the H₂O₂-dependent inhibition of SENP1 and increases overall activity. Note also, in these experiments we used a C608S mutant, which has no effect on SENP1

activity, as control since we felt it important to balance the loss of cysteine thiols across groups. We also present a schematic proposal for how this proton transfer may increase C603 reactive thiolate, and the possible impact of C535 (**Fig. S6C**).

Finally, unrelated to the above comments, but related to general 'regulation of SENP1 activity', I will acknowledge a paper (PMID: 36921622) published during this revision has demonstrated a role for SENP1 in the mediating the regulation of mitosis by lactate. Although not particularly related to this present study (beta-cells are thought to have low levels of lactate, although this is a bit controversial recently: <https://www.biorxiv.org/content/10.1101/2022.12.21.521364v1>), that paper nicely showed that Zn²⁺ inhibits SENP1 activity and binds to SENP1 in complex with lactate at C535. This is consistent with our demonstration of Zn²⁺ regulation of SENP1 (with redox) via C535, and we have now added reference and discussion of this work during revision (page 10).

REVIEWERS' COMMENTS

Reviewer #1 (Remarks to the Author):

The authors have addressed my previous concerns and in my opinion have also satisfactorily addressed those of my fellow reviewers.

Reviewer #2 (Remarks to the Author):

The authors have done a great job and the manuscript has been improved. I especially want to applaud the use of SuperPlots and doing statistics also on mice/donors and not only cells! I only have a few small things that the authors might want to address.

1. Related to comment 6 (and your response) on the previous version of the manuscript: I get that during the starvation before the experiment a larger pool of granules ready to be released accumulate in overweight individuals, but I am still not sure I understand why this does not result in differences also at the higher glucose concentrations in Fig 1b. Have these “extra” granules already been released when you depolarize the membrane? I.e., if you depolarized cell that were just put in 5 or 10mM glucose, would you have a bigger response in overweight non-diabetics? Note that I am not asking you to perform any experiments, maybe just tweak the text some more as I just want to understand and make sure everything is easily understandable for the reader.

2. The pathways you list on page 5 (histone methylation (29), MAPK signaling, cholesterol biosynthesis, insulin receptor signaling, and ER-associated misfolded protein response) are not in Supp table 3. Have the pathways presented in Fig 2b been excluded from the table?

3. I would mention that the RNAseq data is not corrected for multiple testing

Reviewer #3 (Remarks to the Author):

The authors have done a nice job of addressing all reviewer's major concerns. They have made modifications to the manuscript text that add needed clarity and they added new experiments that improve the overall strength and quality of the study. The studies provide

interesting and valuable mechanistic insights into regulation of the SENP1 SUMO protease and its role in insulin secretion.

Response to reviewers:

Reviewer #1 (Remarks to the Author):

The authors have addressed my previous concerns and in my opinion have also satisfactorily addressed those of my fellow reviewers.

We thank the reviewer for their time and effort in the review of our paper.

Reviewer #2 (Remarks to the Author):

The authors have done a great job and the manuscript has been improved. I especially want to applaud the use of SuperPlots and doing statistics also on mice/donors and not only cells! I only have a few small things that the authors might want to address.

We thank the reviewer again for their useful comments and suggestions.

1. Related to comment 6 (and your response) on the previous version of the manuscript: I get that during the starvation before the experiment a larger pool of granules ready to be released accumulate in overweight individuals, but I am still not sure I understand why this does not result in differences also at the higher glucose concentrations in Fig 1b. Have these “extra” granules already been released when you depolarize the membrane? I.e., if you depolarized cell that were just put in 5 or 10mM glucose, would you have a bigger response in overweight non-diabetics? Note that I am not asking you to perform any experiments, maybe just tweak the text some more as I just want to understand and make sure everything is easily understandable for the reader.

We have tried to clarify this further in the manuscript (page 4). In brief, in lean individuals (or chow fed mice) the ‘preparation of granules for release’ is increased at higher glucose...allowing for a robust pool of granules to be secreted when Ca²⁺ rises. In obese individuals (or HFD fed mice), this process appears to occur at BOTH low and high glucose levels, so that the pool of releasable granules is filled regardless of glucose (presumably, there is a maximum extent to which those pools can be filled). Then those granules are released upon Ca²⁺ increase triggered by patch-clamp-depolarization.

2. The pathways you list on page 5 (histone methylation (29), MAPK signaling, cholesterol biosynthesis, insulin receptor signaling, and ER-associated misfolded protein response) are not in Supp table 3. Have the pathways presented in Fig 2b been excluded from the table?

We thank the reviewer for pointing this out, as it brought our attention to an error on our part – we should have included two tables (Suppl Table 3, related to Fig 2B; and Suppl Table 4, related to Fig 2C). We had only included the latter (pointing mistakenly to Fig 2B). We have now fixed this in text (referring to Suppl Table 4, page 5), and including

both tables in the resubmission. The new Suppl Table 3 indeed includes the pathways that the reviewer indicates.

3. I would mention that the RNAseq data is not corrected for multiple testing

We have now mentioned this in the Methods section (page 14).

Reviewer #3 (Remarks to the Author):

The authors have done a nice job of addressing all reviewer's major concerns. They have made modifications to the manuscript text that add needed clarity and they added new experiments that improve the overall strength and quality of the study. The studies provide interesting and valuable mechanistic insights into regulation of the SENP1 SUMO protease and its role in insulin secretion.

We thank the reviewer for these kind comments, and their effort in reviewing this work.